# Extracellular vesicles secreted by *Giardia duodenalis* regulate host cell innate immunity via TLR2 and NLRP3 inflammasome signaling pathways

**Panpan Zhao**[1,2◉], **Lili Cao**[1,3◉], **Xiaocen Wang**[1◉], **Jingquan Dong**[2], **Nan Zhang**[1], **Xin Li**[1], **Jianhua Li**[1], **Xichen Zhang**[1], **Pengtao Gong**[1]*

**1** Key Laboratory of Zoonosis, College of Veterinary Medicine, Jilin University, Changchun, China, **2** Jiangsu Key Laboratory of Marine Biological Resources and Environment, Jiangsu Key Laboratory of Marine Pharmaceutical Compound Screening, Co-Innovation Center of Jiangsu Marine Bio-industry Technology, Jiangsu Ocean University, Lianyungang, China, **3** Jilin Academy of Animal Husbandry and Veterinary Medicine, Changchun, China

◉ These authors contributed equally to this work.
* gongpt@jlu.edu.cn

**Data Availability Statement:** All relevant data are within the manuscript and its Supporting Information files.

## Abstract

*Giardia duodenalis*, also known as *G. intestinalis* or *G. lamblia*, is the major cause of giardiasis leading to diarrheal disease with 280 million people infections annually worldwide. Extracellular vesicles (EVs) have emerged as a ubiquitous mechanism participating in cells communications. The aim of this study is to explore the roles of *G. duodenalis* EVs (GEVs) in host-pathogen interactions using primary mouse peritoneal macrophages as a model. Multiple methods of electron microscopy, nanoparticle tracking analysis, proteomic assays, flow cytometry, immunofluorescence, qPCR, western blot, ELISA, inhibition assays, were used to characterize GEVs, and explore its effects on the host cell innate immunity as well as the underlying mechanism using primary mouse peritoneal macrophages. Results showed that GEVs displayed typical cup-shaped structure with 150 nm in diameter. GEVs could be captured by macrophages and triggered immune response by increasing the production of inflammatory cytokines Il1β, Il6, Il10, Il12, Il17, Ifng, Tnf, Il18, Ccl20 and Cxcl2. Furthermore, activation of TLR2 and NLRP3 inflammasome signaling pathways involved in this process. In addition, CA-074 methyl ester (an inhibitor of cathepsin B) or zVAD-fmk (an inhibitor of pan-caspase) pretreatment entirely diminished these effects triggered by GEVs exposure. Taken together, these findings demonstrated that GEVs could be internalized into mouse peritoneal macrophages and regulate host cell innate immunity via TLR2 and NLRP3 inflammasome signaling pathways.

## Author summary

*G. duodenalis*, one of the most common cause of diarrheal diseases, is widely existed in the contaminated water and threatening the public health especially in developing

**Funding:** PTG received funding from the National Science Foundation of China (No.31772732 and 31101804) and the Fundamental Research Funds for the Central Universities for financial support. XCZ received funding from the National Science Foundation of China (No.31672288) for financial support. The funders had no role in study design, data collection and analysis, decision to publish, or preparation of the manuscript.

**Competing interests:** The authors have declared that no competing interests exist.

countries. Along with the increasing resistance to anti-*G. duodenalis* drugs occurs, new targets against giardiasis are of urgently needed. The innate immune system is the first defense line of organism to resist multiple pathogens invasion through recognizing pathogen-associated molecular patterns (PAMPs) by pattern recognition receptors (PRRs), termed Toll-like receptors (TLRs) on the surface of cell membrane and nucleotide oligomerization domain (Nod)-like receptors (NLRs) inside immune cells. Recently, extracellular vesicles have emerged as a ubiquitous mechanism participating in cells communications. In this study, EVs secreted by extracellular protozoan *G. duodenalis* were obtained and displayed typical cup-shaped structure with 150 nm in diameter. Moreover, GEVs could enter into primary mouse peritoneal macrophages and regulate host cell innate immunity by up-regulation of various inflammatory cytokines expression. Furthermore, TLR2 and NLRP3 inflammasome signaling pathways involved in this process. This study demonstrated that GEVs could be internalized into primary mouse peritoneal macrophages, regulate host cell innate immunity via TLR2 and NLRP3 inflammasome signaling pathways, and may provide new targets against giardiasis.

## Introduction

*Giardia duodenalis*, also known as *G. intestinalis* or *G. lamblia*, assemblages A and B are important zoonotic protozoans leading to diarrheal disease especially for children under five years old in developing countries [1]. *G. duodenalis* is transmitted by the fecal-oral route through ingestion of contaminated water [2,3]. Infectious cysts are uptaken by mouth, entered into stomach and transformed to trophozoites in the intestine going through a complicated process [4]. Giardiasis is endemic worldwide and often occurs in groups of travelers [5]. It is estimated that about 280 million diarrhea infections are caused by giardiasis annually [4]. Considering the huge influence brought by *G. duodenalis*, giardiasis has been added into the Neglected Diseases Initiative by the World Health Organization since 2006 [6,7]. Giardiasis are attracting substantial attention by public. To control and treat with giardiasis, nitazoxanide, metronidazole, and tinidazole are durgs of choice [8,9]. However, the increasing resistance to these anti-giardiasis drugs are common in recently years [10,11]. Hence, it is urgently needed to look for new targets to prevent and treat with giardiasis.

The innate immune system is the first defense line of organism to resist multiple pathogens invasion through triggering non-specific immune response in immune cells. Therefore, an in-depth study of the immune mechanisms that mediate host resist to *G. duodenalis* would help to develop new approaches to control giardiasis. Pattern recognition receptors (PRRs), termed Toll-like receptors (TLRs) and nucleotide oligomerization domain (Nod)-like receptors (NLRs), in innate immune cells, such as macrophages, dendritic cells, could recognize the pathogen-associated molecular patterns (PAMPs) of pathogens [12,13]. TLRs are transmembrane signaling receptors and activation of TLRs not only involves in inflammatory responses but also regulates adaptive immunity. Previous reports indicated that TLR2 involved in the initial recognition of *G. duodenalis* trophozoites, influenced the production of proinflammatory cytokines IL-6, IL-12 and TNF-α in WT mouse peritoneal macrophages, increased the parasite burden in hosts and aggravated giardiasis when comparing with the TLR2 knockout mice [14]. However, the interaction of PAMPs in *G. duodenalis* with the PRRs on innate immune cells has not yet been fully elucidated.

NLRs are intracellular innate immune receptors containing NOD1, NOD2, and NLRP3, etc. NOD1 and NOD2 mediate the activation of NF-κB signal pathway [15]. Other NLRs

could recognize the intracellular pathogens or danger-associated molecular patterns (DAMPs), induce the assembly of the inflammasome. Inflammasomes are multiprotein complexes composed of NLR family members and/or apoptosis-associated speck-like protein (ASC). They could mediate immune response to resist or promote pathogens infection. NLRP3 inflammasome is most well-investigated for it could be widely activated by various particles, uric acid crystals, toxins, bacterials, viruses, as well as parasites through caspase-1 dependent canonical pathway or caspase-11 dependent non-canonical pathway [16–20]. Activation of the NLRP3 inflammasome requires two signals. The first signal is the stimuli triggered NF-κB signal pathway activation leading to the up-regulation of NLRP3, pro-IL-1β and pro-IL-18 mRNA level [21]. The second signal is the stimuli mediated NLRP3 oligomerization with ASC and pro-caspase-1 or pro-caspase-11/4, the activated caspase-1 or caspase-11/4 would cleave pro-IL1β/pro-IL-18 into the mature IL-1β/IL-18 [21]. A newly research found that *G. duodenalis* could promote the production of antimicrobial peptides and attenuate disease severity induced by attaching and effacing enteropathogens via the induction of the NLRP3 inflammasome [22]. However, the unequivocal role of NLRP3 inflammasome in the *G. duodenalis* triggered host innate immunity are poorly understood.

As an extracellular pathogen, limited research reports the immune mechanisms mediated by the intracellular NLRs in *G. duodenalis*. Communications between cells is mediated by biological molecules, including proteins, lipids, and nucleic acids, which are widely existed in extracellular vesicles (EVs) [23–25]. Moyano et al. reported that *G. duodenalis* could release exosome-like vesicles, which was closely related with endosomal sorting complex required for transport-associated AAA+-ATPase Vps4a, Rab11, and ceramide [26]. Varieties of cells could secrete EVs, activate intracellular signal pathways, and trigger physiological responses [27]. Previous reports indicated that extracellular parasite *Trichomonas vaginalis* could modulate of the innate immunity of the host cell reducing the inflammatory chemokines IL-8 expression in the host ectocervical cells by increasing EVs secretion [28]. Extracellular parasite *Trypanosoma brucei* responded to innate immunity by developing evasion mechanisms of extracellular vesicle release [29]. Gavinho et al. reported that *G. duodenalis* could shed large EVs and small EVs, which contained different protein contents and affected host-pathogen interactions by hindering adhesion to host cells [30]. Evans-Osses et al. reported that microvesicles released from *G. duodenalis* could be captured by human immature dendritic cells, increased the activation and allostimulation of human dendritic cells, played roles in the attachment to Caco-2 cells [31]. The present study aims to explore the roles of EVs secreted by *G. duodenalis* trophozoites in host-pathogen interactions using primary mouse peritoneal macrophages as a model.

In the present study, we characterized *G. duodenalis* EVs (GEVs), verified GEVs could regulate immune response by triggering inflammatory cytokines transcription and secretion in hosts, investigated the roles of TLR2 and NLRP3 inflammasome signaling pathway in GEVs induced inflammatory response in murine macrophages and revealed the underlying mechanisms in *G. duodenalis*-host interactions.

## Methods

### Ethics statement

All animal experiments have received approval for research ethics from the Animal Welfare and Research Ethics Committee of Jilin University and the certificate number is pzpx20190929065. Six-eight weeks old C57BL/6 female mice were maintained in feeding cages with sterile food, water, and a 12 h light/dark cycle. The mice were used for experiments after an acclimatization period of more than 7 days and euthanized before isolation of peritoneal macrophages.

### *G. duodenalis* and GEVs preparation

Trophozoites of *G. duodenalis* WB strain (clone C6, ATCC30957; American Type Culture Collection, Manassas, VA, USA) were grown in sterilized modified TYI-S-33 medium with 12.5% fetal bovine serum (Every Green, Zhejiang), 0.1% bovine bile (Sigma, USA), 50 mg/mL Gentamicin sulfate, 100 U/mL Penicillin, and 100 μg/mL Streptomycin sulfate (Biological Industries, Israel) at 37˚C under microaerophilic conditions. *G. duodenalis* was subcultured when confluent. To collect *G. duodenalis*, the culturing medium was replaced with fresh medium to eliminate dead parasites. The tubes were placed on ice for 20 min and *G. duodenalis* was collected by centrifugation (2,000 × g for 8 min at 4˚C). After washing three times with PBS, *G. duodenalis* was counted and diluted to $1 \times 10^8$ parasites/mL.

GEVs were enriched and purified as previously described with a little modification to the protocol [26,31,32]. The *G. duodenalis* trophozoites were diluted into $1 \times 10^6$ parasites/mL in modified TYI-S-33 medium with 12.5% exosome-depleted fetal bovine serum (Biological Industries, Israel) and cultured at 37˚C for 12 h [33]. The culturing supernatants were collected and centrifuged at 2,000 × g for 10 min at 4˚C to remove *G. duodenalis*. Then, supernatants were collected and centrifuged at 10,000 × g for 45 min. Following, the supernatants were filtered through a 0.22 μm sterilized PES membrane (Merck Millipore, USA) and then ultracentrifuged at 100,000 × g for 60 min at 4˚C on a ultracentrifuge (Hitachi, Japan). The collected GEVs were washed twice in 5.5 mL sterilized PBS, resuspended in 200 μL PBS and quantified with a BCA Protein Assay Kit (Thermo Scientific, USA).

### Transmission electron microscopy observation

For morphology observation, 10 μL of GEVs were immediately added to a carbon-coated copper grid and remained for 1 min at RT. Then, the GEVs were negatively stained with 20 μL of 3% phosphotungstic acid for 5 min. After removal of the redundant liquid, GEVs samples were observed on a transmission electron microscopy (TEM, HITACHI, Japan).

### Nanoparticle tracking analysis

The particle size and number of GEVs were measured using Nanoparticle tracking analysis (NTA). In detail, fresh GEVs were diluted to $1 \times 10^7$ particles/mL in PBS, subjected to a Zeta-View PMX 110 instrument (Particle Metrix, Germany), imaged 30 frames per second and lasted for 1 min. Samples were measured in triplicate at 25˚C. Data was analyzed using Zeta-View 8.02.28 software.

### Proteomic assays

GEVs samples were fractionated into soluble fractions, analyzed using SDS-PAGE, and digested with trypsin. LC-MS/MS assays were carried out as previous described [32]. Peptides were analyzed on an Orbitrap-Elite mass spectrometer (Thermo Scientific, USA). The MS/MS data were searched against the uniprot *G. duodenalis* WB C6 strain database with the Mascot Proteome Discoverer 2.3 (Thermo Scientific, USA). Data was submitted to the Mascot server through Proteome Discoverer and set the search parameters as follows: single isotope mass; trypsin digestion, no more than 2 missed cutting sites, 2+, 3+ and 4+ charges in the peptide; fixed modification was carbamidomethylation (C); dynamically modification was oxidation (M) and Acetthyl (Protein N-term); the maximum error of the parent ion was 20 ppm; the maximum error of the fragment ion was 0.1Da. The peptides were exported when FDR≤0.01 (high confidence).

### Cell culture and PKH67-labeled GEVs

To enrich macrophages, mice were inoculated intraperitoneally with 2.5 mL of 2.98% Difo Fluid thioglycollate medium (BD, USA) for 3.5 days and macrophages were harvested from the peritoneal cavity. After washing twice with PBS, peritoneal macrophages were resuspended in RPMI 1640 medium with 10% fetal bovine serum (Biological Industries, Israel), diluted into $1.5 \times 10^6$ cells/mL, and cultured at 37˚C/5% $CO_2$ in 6-well plates at $4.5 \times 10^6$ cells/well, 12-well plates at $1.5 \times 10^6$ cells/well or 24-well plates at $5 \times 10^5$ cells/well. The cell purity was verified using flow cytometry through staining with APC anti-mouse/human CD11b (1:200, BioLegend, USA).

To determine whether GEVs could enter into mouse peritoneal macrophages, GEVs were stained using a PKH67 Green Fluorescent Cell Linker Kit (Sigma, St. Louis, MO, USA) [31,32]. In detail, 50 µg of GEVs were dissolved in 100 µL of PBS and mixed with 1 mL Diluent C and 4 µL of PKH67 dye. The mixture was incubated for 4 min at room temperature (RT) in darkness and then added into 1 mL of 1% BSA to remove excess dye. The PKH67 labeled GEVs were washed with 5.5 mL PBS, ultracentrifuged at 100,000 × g for 1 h, and resuspended in 50 µL PBS.

### GEVs capture assays

The labeled GEVs (25 µg or 12.5 µg) were added into peritoneal macrophages coated in 6-well plates and incubated for 1 h, 3 h, and 6 h at 37˚C. Then, cells were washed in PBS and quantified analysis using flow cytometry. For the inhibition assays, macrophages previously prepared on coverslips in 24-well plates were pretreated with cytochalasin D (10 µM, abcam, USA) at 37˚C for 4 h and then inoculated with 12.5 µg or 25 µg of labeled GEVs for 6 h. Then, cells were washed three times with PBS, fixed in 4% paraformaldehyde at RT for 10 min, permeabilized with 0.1% Triton X-100 at RT for 20 min, stained with 100 nM TRITC Phalloidin (Yeasen, Shanghai) and 10 µg/mL Hoechst 33258 (US EVERBRIGHT, Suzhou). Cells were viewed on a fluorescence microscope (Olympus, Japan).

### Stimulation and noncontact culture system

Murine peritoneal macrophages were stimulated with GEVs (25 µg/mL or 12.5 µg/mL) for 12 h or 24 h. Equal volume PBS and equal amount *G. duodenalis* were used as control. For the *G. duodenalis* control group, a noncontact culture system was set up as previously described with a little modification to the protocol [34]. The transwell insert (Thermo Scientific, USA) with a membrane pore size of 0.4 micron was placed in cell plates coated with murine peritoneal macrophages, and *G. duodenalis* were added into transwell inserts culturing in RPMI 1640 medium with 2% fetal bovine serum.

### RNA extraction and real-time quantitative PCR analysis

Total RNA was extracted from the infected murine peritoneal macrophages coated on 12-well plate using TRIzol reagent (Monad, Wuhan) according to the manufactory's instructions. RNA quality was evaluated by measuring parameters of concentration and purity using Nanodrop ND-2000 apparatus (Thermo Scientific, USA). cDNA was synthesized using 0.5–2 µg total RNA after removal of gDNA in the total RNA with MonScript RTIII Super Mix with dsDNase (Two-Step) (Monad, Wuhan). The reverse transcription reaction contained 9 µL of RNA template, 4 µL of 5 × RTIII Super Mix, and 7 µL of nuclease-free water. The reaction conditions were set as follows: reverse transcription at 37˚C for 30 min and inactivation at 85˚C for 3 min.

**Table 1.  Primers used for qRT-PCR analysis of cytokines.**

| Gene | Genbank number | Primer sequence (5'to 3') | Product size (bp) | Primer length (nt) | Cross intron length (nt) | Primer site |
|------|----------------|---------------------------|-------------------|---------------------|--------------------------|-------------|
| Il12 | MMU23922 | TACAAGGTTCAGGTGCGAGC | 158 | 20 | 0 | 1667...1686 |
|      |          | ATGTATCCGAGACTGCCCAC |     | 20 |   | 1824...1805 |
| Il6 | NC_000071 | TGCCTTCTTGGGACTGATGC | 216 | 20 | 1272 | 279...298 |
|     |           | GCAAGTGCATCATCGTTGTTC |     | 21 |      | 1765...1745 |
| Il10 | NC_000067 | GCAGTGGAGCAGGTGAAGAG | 250 | 20 | 249 | 2678...2697 |
|      |           | CGGAGAGAGGTACAAACGAGG |     | 21 |     | 4548...4528 |
| Il17a | NM_010552.3 | AACATGAGTCCAGGGAGAGC | 225 | 20 | 1108 | 55...74 |
|       |             | ACGTGGAACGGTTGAGGTAG |     | 20 |      | 280...261 |
| Tnf | NM_013693 | GACGTGGAACTGGCAGAAGA | 253 | 20 | 696 | 192...211 |
|     |           | GGCTACAGGCTTGTCACTCG |     | 20 |     | 446...427 |
| Ifng | NM_008337 | CGGCACAGTCATTGAAAGCC | 203 | 20 | 10392 | 178...197 |
|      |           | TGTTGTTGCTGATGGCCTGA |     | 20 |       | 380...361 |
| Il1b | NM_008361 | AGGAGAACCAAGCAACGACA | 241 | 20 | 1545 | 582...601 |
|      |           | CTCTGCTTGTGAGGTGCTGA |     | 20 |      | 822...803 |
| Il18 | NM_008360 | ACCAAGTTCTCTTCGTTGAC | 149 | 20 | 2070 | 747...766 |
|      |           | CTTCACAGAGAGGGTCACAG |     | 22 |      | 895...876 |
| Ccl20 | NM_001159738 | CGTCTGCTCTTCCTTGCTTTG | 219 | 21 | 985 | 77...97 |
|       |              | CTGCTTTGGATCAGCGCAC |     | 19 |     | 277...295 |
| Cxcl2 | NM_009140 | CTGGCCACCAACCACCAG | 180 | 18 | 213 | 101...118 |
|       |           | GCAAACTTTTTGACCGCCCT |     | 20 |     | 261...280 |
| Actb | NM_007393 | GCCATGTACGTAGCCATCCA | 240 | 20 | 455 | 391...410 |
|      |           | ACGCACGATTTCCCTCTCAG |     | 20 |     | 630...611 |

Primers of inflammatory cytokines and pattern recognition receptors were designed and listed in Tables 1–3. Primer specificity was verified by analyzing the melting curve. Real-time quantitative PCR (qPCR) was performed using MonAmp SYBR Green qPCR Mix (None ROX, Monad, Wuhan) on a LightCycler 480 II machine (Roche, Germany). The qPCR reaction contained 1 μL of 20 times diluted cDNA template, 10 μL of MonAmp SYBR Green qPCR Mix, 4 μL of forward and reverse primer (1 μM), and 5 μL of nuclease-free water. The qPCR reaction conditions were set as follows: denaturation at 95˚C for 30 s, followed by 40 cycles of 95˚C for 10 s and 60˚C for 30 s, and the melting curve was set as the default. Results were normalized to expression of the housekeeping gene actin, and the relative mRNA fold change was calculated as $2^{-\Delta\Delta Ct}$, where ΔCt represents the Ct (target gene)—Ct (actin) and ΔΔCt represents the ΔCt (sample)—ΔCt (control).

## Enzyme linked immunosorbent assays

Inflammatory cytokines in supernatants were measured using Mouse IL-1 beta, IL-6, and TNF-alpha ELISA Kit (Invitrogen, USA) according to the manufacturer's instructions. In detail, capture antibodies were coated on the ELISA plate at 4˚C overnight and blocked at 37˚C for 1 h after washing three times. Standard samples and supernatants were added into plates and incubated at 4˚C overnight. ELISA Diluent was used as blank control. Detection antibodies were added into plates after washing five times and incubated at 37˚C for 1 h. Streptavidin-HRP was added into plates and incubated at 37˚C for 0.5 h. Then, TMB solution was added after washing five times and incubated at 37˚C for 15 min. Reaction was stopped by adding stop solution and absorbance at 450 nm were measured using microplate reader. The obtained OD values were converted to pg/mL by interpolation of the standard curve.

**Table 2. Primers used for qRT-PCR analysis of TLRs.**

| Gene | Genbank number | Primer sequence (5'to 3') | Product size (bp) | Primer length (nt) | Cross intron length (nt) | Primer site |
|---|---|---|---|---|---|---|
| TLR1 | NC_000071 | AGTCAGCCTCAAGCATTTGG | 115 | 20 | 0 | 6716…6735 |
| | | TACCCGAGAACCGCTCAAC | | 19 | | 6830…6812 |
| TLR2 | NC_000069 | CGCTCCAGGTCTTTCACCTC | 101 | 20 | 0 | 3165…3184 |
| | | AGGTCACCATGGCCAATGTA | | 20 | | 3265…3246 |
| TLR3 | NC_000074 | CGCAGTTCAGCAAGCTATTG | 103 | 20 | 0 | 13950…13969 |
| | | TCTTCGCAAACAGAGTGCAT | | 20 | | 14052…14033 |
| TLR4 | NC_000070 | ACTGTTCTTCTCCTGCCTGACA | 99 | 22 | 5974 | 323…344 |
| | | GGACTTTGCTGAGTTTCTGATCC | | 23 | | 6395…6373 |
| TLR5 | NC_000067 | GGATGCTGAGTTCCCCCAC | 134 | 19 | 700 | 17686…17704 |
| | | AAAGGCTATCCTGCCGTCTG | | 20 | | 18519…18500 |
| TLR6 | NC_000071 | TCCGACAACTGGATCTGCTC | 101 | 20 | 0 | 15886…15905 |
| | | AAGACTTTCTGTTTCCCCGC | | 20 | | 15986…15967 |
| TLR7 | NC_000086 | CCGTTGAGAGAGTTGCGGTA | 187 | 20 | 0 | 23707…23726 |
| | | TGAGTTTGTCCAGAAGCCGTA | | 21 | | 23893…23873 |
| TLR8 | NC_000086 | CTGACGTGCTTTTGTCTGCTG | 101 | 21 | 0 | 18508…18528 |
| | | AGGGAGTTGTGCCTTATCTCGT | | 22 | | 18608…18587 |
| TLR9 | NC_000075 | CTGCCCAAACTCCACACTCT | 100 | 20 | 0 | 2082…2101 |
| | | ACAAGTCCACAAAGCGAAGG | | 20 | | 2181…2162 |
| TLR11 | NC_000080 | TTGGGATTGGAAATGACAGG | 173 | 20 | 0 | 5104…5123 |
| | | CAACAGCAGGAGATGAGTGG | | 20 | | 5276…5257 |
| TLR12 | NC_000070 | TAACTGGGTGGAGCACTTCC | 143 | 20 | 0 | 214…233 |
| | | CAAGGTCTGTGTCAGGTTGC | | 20 | | 356…337 |
| TLR13 | NC_000086 | AAAGACACGGGATTCAGGTTG | 102 | 21 | 0 | 15192…15212 |
| | | GGTGGTCCAGGAATACAGAGG | | 21 | | 15293…15273 |

**Table 3. Primers used for analysis of NLRs.**

| Gene | Genbank number | Primer sequence (5'to 3') | Product size (bp) | Primer length (nt) | Cross intron length (nt) | Primer site |
|---|---|---|---|---|---|---|
| NOD1 | NM_172729.3 | F: GATTGGAGACGAAGGGGCAA | 223 | 20 | 1835 | 2934…2953 |
| | | R: CGTCTGGTTCACTCTCAGCA | | 20 | | 3156…3137 |
| NOD2 | NM_145857.2 | F: GCCAGTACGAGTGTGAGGAG | 218 | 20 | 10262 | 478…497 |
| | | R: GCGAGACTGAGTCAACACCA | | 20 | | 695…676 |
| NLRP1 | NC_000077 | F: ATAAACAAGCCACCCCCAGT | 152 | 20 | 1602 | 46013…46032 |
| | | R: TGTGCCCAATGTCGATCTCA | | 20 | | 47766…47747 |
| NLRP2 | NC_000073 | F: AGGCGGTCTTTCCAGAGAATG | 173 | 21 | 2345 | 23574…23594 |
| | | R: TCCAGTGCAGAGCTGTTGAG | | 20 | | 26091…26072 |
| NLRP3 | NM_145827.4 | F: AGCCAGAGTGGAATGACACG | 230 | 20 | 4488 | 492…511 |
| | | R: CGTGTAGCGACTGTTGAGGT | | 20 | | 721…702 |
| NLRP6 | NM_133946.2 | F: TTGTTCGACAGGCTCTCAGC | 226 | 20 | 961 | 2045…2064 |
| | | R: ACTGGGGGTTGTTTCTTGGT | | 20 | | 2270…2251 |
| NLRP12 | NM_001033431 | F: GTCTGCTCGTTTTGTGCGAG | 235 | 20 | 4507 | 2125…2144 |
| | | R: TGCCCAAGGCATTTCGGTAT | | 20 | | 2359…2340 |
| NLRC4 | NM_001033367 | F: GCTCAGTCCTCAGAACCTGC | 165 | 20 | 3145 | 13872…13891 |
| | | R: ACCCAAGCTGTCAATCAGACC | | 21 | | 17181…17161 |
| NLRC5 | NM_001033207 | F: TCTCTAAGCAGCTAGGGGCA | 220 | 20 | 2544 | 296…315 |
| | | R: GGGGAGTGAGGAGTAAGCCA | | 20 | | 515…496 |

## Protein extraction, SDS-PAGE, and western blot

Protein samples from supernatants were prepared as previously described [19]. Cell culture supernatants in 6-well plate were precipitated by gently blending with equal volume cold methanol and a quarter volume cold chloroform and centrifuged at 13,000× g for 10 min at 4˚C. Next, the upper phase was discarded, and protein were rinsed in 500 μL of cold methanol. After centrifugation at 13,000 × g for 10 min, pellets were dried at RT and dissolved in 30 μL of 1% SDS solution. Protein samples from cells were extracted using RIPA buffer containing 1 mM PMSF (Solarbio, Beijing) according to the manufactory's instructions. Briefly, cells were scraped from plates, treated with 200 μL of RIPA buffer, and fully lysated at 4˚C for 5 min. Then, cell lysates were centrifuged at 13,000 × g for 10 min at 4˚C and concentrated using methanol-chloroform method. Protein concentrations were measured using BCA Protein Assay Kit (Thermo Scientific, USA). Protein samples were mixed with 6 × Protein Loading Buffer (TransGen, Beijing), boiled for 10 min and stored at -80˚C.

Thirty microgram of protein samples were separated on a 12% SDS-PAGE by electrophoresis in Tris-Glycine-SDS (TGS) buffer under conditions of 80 V for 1 h and then 120 V for 30 or 60 min. Next, the protein samples were transferred into 0.22 or 0.45 μm PVDF membrane (Millipore, USA) under conditions of 200 mA for 1 h or 2 h. Membranes were blocked in 5% fish gelatin protein at room temperature (RT) for 2 h, incubated with primary antibodies of IL-1β (R&D, USA), caspase-1 (p20) (Adipogen, Switzerland), NLRP3 (Adipogen, Switzerland) and β-actin (Proteintech, Wuhan) overnight at 4˚C. After washing three times with PBST, membranes were incubated with secondary antibodies of HRP-conjugated rabbit anti-goat IgG (H+L) and goat anti-mouse IgG (H+L) (Earthox, USA). After washing three times with PBST, the membranes were detected with Immobilon Western Chemiluminescent HRP substrate (Millipore, USA) and blots were visualized on a ChemiScope Western Blot Imaging System (Clinx, Shanghai).

## Inhibition assays

To monitor the roles of activated TLR and NLR in response to GEVs and *G.duodenalis* infection in murine peritoneal macrophages, cells were pretreated with 50 μM C29 (an inhibitor of TLR2 by blocking hTLR2/1 and hTLR 2/6 signal, MedChemExpress, USA), 50 μM Glibenclimide (an inhibitor of NLRP3 by inhibiting K$^+$ efflux; Selleck, Shanghai), or 25 μM CA-074 methyl ester (CA-074 Me, an inhibitor of cathepsin B, MedChemExpress, USA) for 1 h before stimulation. The mRNA fold change of activated TLR and NLR and downstream inflammatory cytokines were determined using qPCR, and inflammatory cytokines protein levels were determined using ELISA and western blot.

To monitor the role of activated inflammasome in response to GEVs and *G. duodenalis* infection in murine peritoneal macrophages, cells were pre-treated with 50 μM Glibenclimide, 25 μM CA-074 methyl ester, 100 μM Ac-YVAD-CHO (an inhibitor of caspase-1 and -4; Enzo Life Science, Switzerland), or 10 μM zVAD-fmk (an inhibitor of pan-caspase; Selleck, Shanghai) for 1 h before stimulation. The mRNA fold change of activated NLR/caspase-1 and downstream inflammatory molecules were determined using qPCR, and inflammatory molecules protein levels were determined using ELISA and western blot.

## Immunofluorescence staining

Immunofluorescence assays were carried out to determine whether the up-regulated TLR and NLRP receptors were activated. Macrophages previously prepared on coverslips in 24-well plates were inoculated with 25 μg/mL of GEVs for 12 h, fixed in 4% paraformaldehyde at RT for 10 min, and permeabilized with 0.1% Triton X-100 at RT for 20 min. Then, cells were

blocked in 5% BSA at RT for 2 h and incubated with primary antibodies of NLRP3 (1:100) or TLR2 (1:100, ABclonal, Wuhan) at 4˚C overnight. Next, cells were incubated with secondary antibodies of FITC-conjugated goat anti-mouse IgG (H+L) or Cy3-conjugated goat anti-rabbit IgG (H+L) (1:400, Earthox, USA) for 1 h at 37˚C. Nuclei in the cells were stained with 10 μg/mL Hoechst 33258. Cells were viewed on a fluorescence microscope (Olympus, Japan).

## Statistical analysis

Differences between two groups were analyzed by t-test and multiple groups were used analyzed by one-way ANOVAusing SPSS 22.0 (Chicago, IL). Results were expressed as the mean ± SEM. The homogeneity of variance of data are analyzed by Levene test. Data of homogeneity of variance were further analyzed using post-test of bonferroni(B) and heterogeneity of variance were further analyzed using Tamhane's T2(M). The graphs were generated in Graph-Pad Prism 7.00 (Inc., La Jolla, USA). All experiments were performed three times with three technical replicates. Significance is shown as $^*p < 0.05$, $^{**}p < 0.01$, and $^{***}p < 0.001$. Three biological replicates were set for each treatment.

## Results

### Characterization and proteomic analysis of GEVs

To verify that GEVs were successfully isolated from *G. duodenalis* trophozoites, the morphological characteristics was determined by TEM negative staining. Results showed that GEVs displayed typical rounded or cup-shaped with membrane bilayer and the diameter was among 100~200 nm (Fig 1A). To further explore the size distribution and concentrations of GEVs,

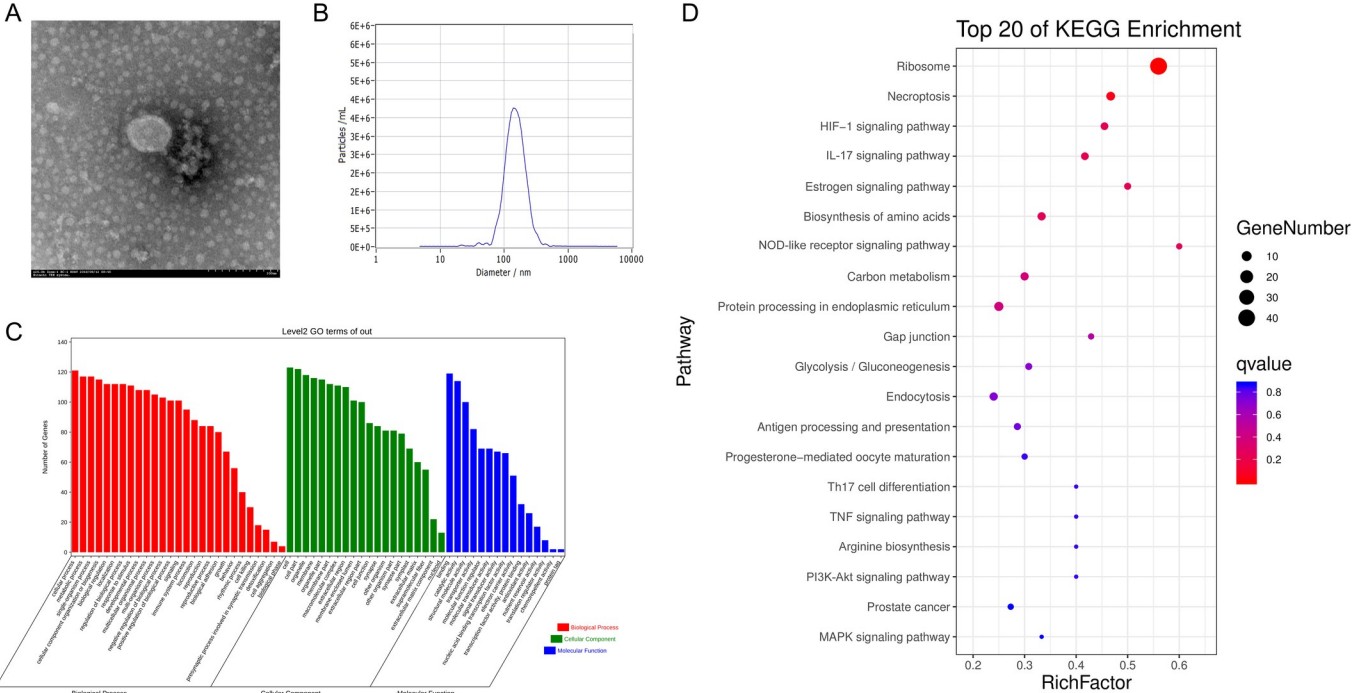

**Fig 1. Characterization and proteomic analysis of *G. duodenalis* extracellular vesicles.** (A) Morphological observation of GEVs using negative staining by TEM. Typical rounded or cup-shaped vesicles were obtained. Scale bar: 200 nm. (B) Nanoparticle tracking analysis of purified GEVs. The concentration and diameter detection of GEVs using Nanosight. The mean diameter and concentration were measured. (C) Gene ontology annotations for all the identified proteins in GEVs. (D) Top 20 of KEGG enrichment in GEVs.

nanoparticle tracking analysis was used to measure millions of GEVs and results showed that the mean diameter at X50 was 143.5 nm and concentration was $4.7 \times 10^{10}$ particles/mL (Fig 1B).

The contents of GEVs were determined by LC-MS/MS assays and results showed that a total of 154 proteins were identified from the trophozoite stage with 49.4% (76/154) proteins containing at least two unique peptides (S1 Table). Among them, 23 proteins were uncharacterized protein, and from the 131 remaining, 26 proteins and 46 proteins individually corresponded to previously identified in *G. duodenalis* microvesicles [31] and large or small extracellular vesicles [30]. Proteins that involved in EVs biogenesis were existed in the GEVs, such as 14-3-3, alpha-tublin, heat shock proteins [32]. Proteins that involved in pathogenesis were overlaps with that identified in microvesicles, large and small extracellular vesicles in *G. duodenalis*, such as VSP, giardin, Arginine deiminase, Ornithine transcarbamylase, etc [30,31]. Gene ontology (GO) annotations showed that 95 genes were related to immune response, 30 genes were associated with cell killing, 82 genes involved in transporter activity, etc. (S2 Table and Fig 1C). The identified proteins may take part in 129 pathways, including Ribosome, Glycolysis, NOD-like receptor signaling pathway, MAPK signaling pathway, PI3K-Akt signaling pathway, Toll-like receptor signaling pathway, mTOR signaling pathway, etc. (S3 Table and Fig 1D). Altogether, these data verified that we successfully isolated GEVs from *G. duodenalis* trophozoites.

## GEVs communicate with mouse peritoneal macrophages and trigger inflammatory response

Previous research have reported that *G. duodenalis* microvesicles could be captured by dendritic cells, moreover, both *G. duodenalis* large and small extracellular vesicles could be internalized by Caco-2 cells [30,31]. To explore whether GEVs could enter murine peritoneal macrophages, GEVs were labeled with lipophilic fluorescent dye of PKH67, which specially labeled the outer membrane of GEVs [35], then incubated into peritoneal macrophages for 1 h, 3 h, and 6 h, and examined using flow cytometry. As shown in Fig 2A, PKH67-labeled GEVs entered into macrophages in a time-dependent manner. GEVs were rapidly internalized into hosts with 25.1% PKH67$^+$ cells at 1 h, 79.2% PKH67$^+$ cells at 3 h, and 89.4% PKH67$^+$ cells at 6 h. No GEVs treated cells were used as negative control. Moreover, different doses of GEVs (25 μg or 12.5 μg) were inoculated into macrophages and results showed that GEVs entered into macrophages in a dose-dependent manner. The amount of PKH67$^+$ cells in the GEVs-treated groups were significantly larger than that in the No GEVs-treated groups (Fig 2B). Inhibition assays showed that the mean gray value of PKH67 in the cytochalasin D-pretreatment groups were lower than that in the untreated group (Fig 2C and 2D). This indicated that GEVs may probably be captured by macrophages through active phagocytosis.

Previous studies have showed that *G. duodenalis* or secreted proteins exposure *in vitro* can not only increase the inflammatory response in epithelial cells [36,37] but also in immune cells, such as macrophages [14]. However, the exact interaction of GEVs with macrophages has not yet been fully elucidated. Then, GEVs were inoculated into peritoneal macrophages coated on 12-well plate to a final concentration of 12.5 μg/mL or 25 μg/mL for 12 h, 18 h, and 24 h, respectively. *G.duodenalis* ($1.5 \times 10^6$ parasites/mL) treated groups were used as positive control. Equal volume of PBS was used as negative control. Cells at the 12 h treatment time-point were collected to determine the mRNA fold change of inflammatory cytokines using qPCR method. Results showed that the mRNA level of Il6, Il10, Ifng, Tnf, Il1β, and Ccl20 displayed a significant increase ($^{***}p < 0.001$), the Il18 and Cxcl2 level significantly up-regulated ($^{**}p < 0.01$) and IL-12 increased ($^*p < 0.05$) when comparing with the PBS-treated control

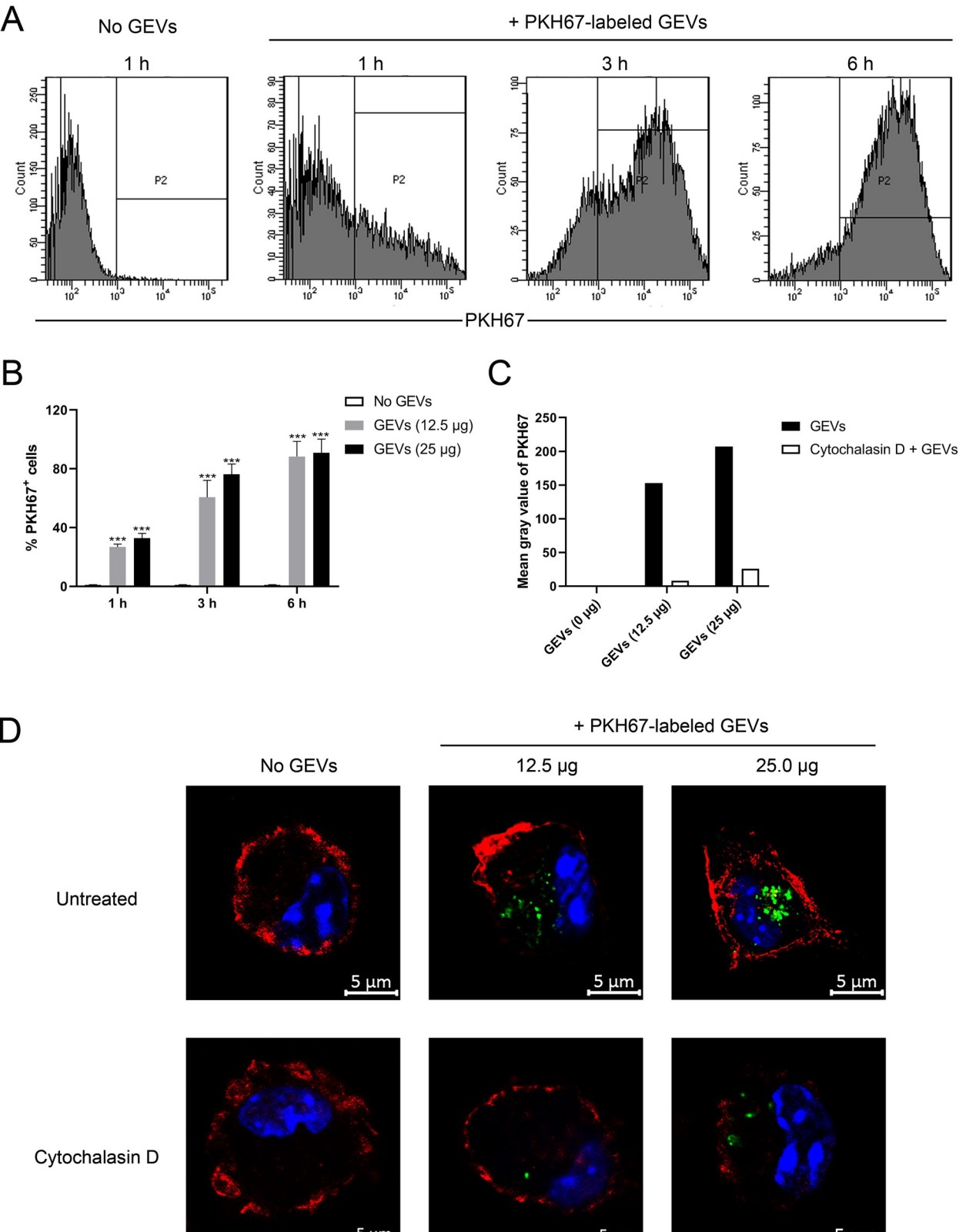

**Fig 2. GEVs are actively captured by murine peritoneal macrophages.** (A) Twenty-five microgram of GEVs were inoculated into murine peritoneal macrophages, incubated for time periods of 1, 3 and 6 h, and detected through flow cytometry. (B) Different doses of GEVs (25 μg or

12.5 µg) were inoculated into macrophages, incubated for the indicated time, and detected through flow cytometry. Significance is shown as ***$p < 0.001$. Three biological replicates were set for each treatment. (C, D) Macrophages were pretreated with 10 µM cytochalasin D for 4 h, inoculated with 12.5 µg or 25 µg of GEVs, and incubated for 6 h. Cells without cytochalasin D pretreatment were used as positive control. Then, cells were stained for confocal microscopy observation: green, PKH67-labeled GEVs; red, host F-actin; blue, nuclei. Scale bars: 5 µm. The mean gray value of PKH67 in GEVs treatment or cytochalasin D combined GEVs treatment groups were calculated using Image J software.

group (Fig 3A). Next, cell culturing supernatants were collected after treatment for 18 h and 24 h, inflammatory cytokines of TNF-α, IL-6, and IL-1β were further determined using ELISA assays. For TNF-α, the contents in the control groups were within 30 pg/mL, in contrast, the GEVs-treated group significantly increased to 1237.25±257.18 pg/mL or 2249.77±188.15 pg/mL at 18 h and 1408.36±179.05 pg/mL or 2605.09±139.32 pg/mL at 24 h in a dose and time-dependent manner (**$p < 0.01$ or ***$p < 0.001$). For IL-6, the contents in the control groups were within 20 pg/mL, in contrast, the GEVs-treated group significantly increased to 211.41±26.34 pg/mL or 387.10±36.72 pg/mL at 18 h and 508.61±53.10 pg/mL or 760.00±96.44 pg/mL at 24 h in a dose and time-dependent manner (***$p < 0.001$). IL-1β was not detected in the control group and the concentrations were up-regulated to 232.16±34.43 pg/mL or 367.68±45.15 pg/mL at 18 h and 292.37±29.88 pg/mL or 478.12±21.46 pg/mL at 24 h in a dose and time-dependent manner (***$p < 0.001$) (Fig 3B). Overall, GEVs triggers these inflammatory cytokines changes both in the transcription levels and protein secretion levels and the significantly increased levels indicates that GEVs play important roles in *G.duodenalis*-induced inflammatory response in macrophages.

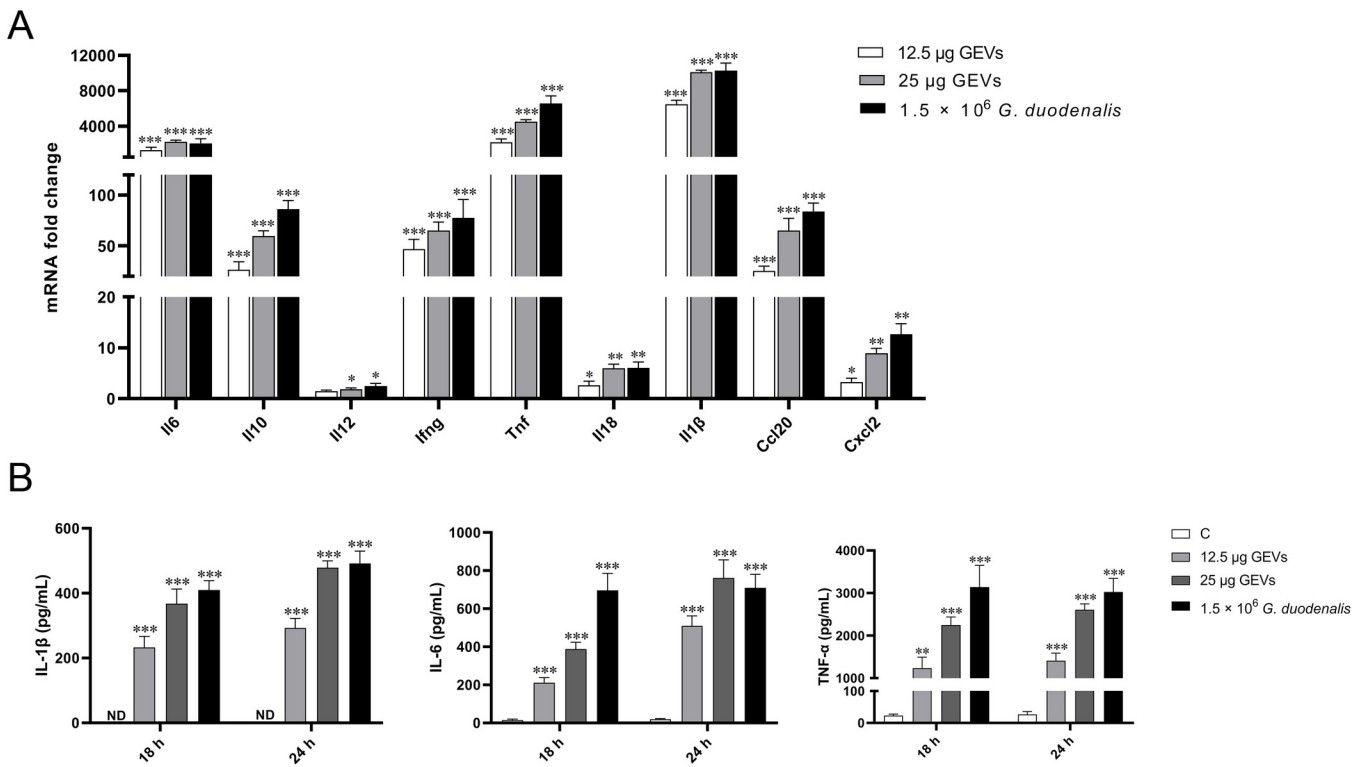

**Fig 3. GEVs induced inflammatory cytokines transcription and secretion in murine peritoneal macrophages.** (A) Cells were incubated with 12.5 µg/mL or 25 µg/mL of GEVs for 12 h, the mRNA levels of inflammatory cytokines, including Il6, Il10, Il12, Ifng, Tnf, Il18, Il1β, Ccl20 and Cxcl2 in cells was measured using qPCR assays. (B) Cells were were incubated with 12.5 µg/mL or 25 µg/mL of GEVs for 18 h and 24 h, supernatants were collected for TNF-α, IL-6, and IL-1β determination using ELISA assays. *G.duodenalis* ($1.5 \times 10^6$ parasites/mL) treated groups were used as positive control. Equal volume PBS-treated group was used as negative control. All experiments were performed three times with three technical replicates and data are mean±SEM. *$p < 0.05$, **$p < 0.01$ or ***$p < 0.001$ vs. control.

### Activation of vital TLR2 and NLRP3 pattern recognition receptors in GEVs inoculated murine peritoneal macrophages

To determine the activated pattern recognition receptors involved in the inflammatory response, a final concentration of 12.5 μg/mL or 25 μg/mL GEVs and $1.5 \times 10^6$ parasite/mL *G. duodenalis* were individually inoculated into murine peritoneal macrophages coated on 12-well plate for 12 h and the mRNA fold change of TLRs in macrophages was measured using qPCR assays. As shown in Fig 4A, the mRNA levels of NOD2, NLRP3, NLRC4 and NLRC5 were significantly increased both in the GEVs treated groups and *G. duodenalis* treated group (**$p < 0.01$ or ***$p < 0.001$). In addition, GEVs could also significantly up-regulated NOD1, NLRP2, and NLRP6 (**$p < 0.01$), however, *G. duodenalis* treated group stayed about the same ($p > 0.05$). Both GEVs and *G. duodenalis* could down-regulate the mRNA levels of NLRP1 and NLRP12 (*$p < 0.05$ or **$p < 0.01$). As shown in Fig 4B, the mRNA levels of TLR2 and TLR7 were significantly increased when treated with either doses of GEVs or *G. duodenalis* comparing with the control group (**$p < 0.01$ or ***$p < 0.001$). The mRNA level of TLR6 was only significantly increased in the high dose of GEVs or *G. duodenalis*-treated groups (*$p < 0.05$ or **$p < 0.01$); in contrast, low dose of GEVs up-regulated the mRNA level of TLR6 though not significant ($p > 0.05$). The mRNA level of TLR4 was significantly up-regulated only in the *G. duodenalis*-treated group (*$p < 0.05$), however, no significant change was observed in GEVs treated groups. Other TLRs, including TLR1, TLR3, TLR9, TLR11, and TLR12, displayed no significant changes between the control group and the experiment groups ($p > 0.05$). For TLR8, the mRNA level was both decreased in these three treated groups vs control ($p > 0.05$ or *$p < 0.05$). For TLR5 and TLR13, the mRNA fold changes were both significantly down-regulated only in the *G. duodenalis* treated group (*$p < 0.05$), however, no significantly change in the GEVs treated groups ($p > 0.05$). Overall, many TLRs and NLRs are up-regulated after treatment with GEVs or *G. duodenalis* and pattern recognition receptors of TLR2 and NLRP3 are the most up-regulated sensor proteins.

To further determine whether the up-regulated TLR2 and NLRP3 pattern recognition receptors were activated, immunofluorescence assays were carried out to locate TLR2 and NLRP3 proteins. NLRP3 was localized to the perinuclear space in puncta in GEV or *G. duodenalis* treated macrophages, in contrast, no FITC-labeled green NLRP3 signals were detected in the No GEVs-treated control group. In No GEVs-treated macrophages, weak red signals (cy3-labeled TLR2 protein) were located in the cell membrane, whereas TLR2 was obviously activated displaying strong red signals in the macrophages incubated with GEV or *G. duodenalis* (Fig 4C). These data indicated that both TLR2 and NLRP3 pattern recognition receptors were activated after treatment with GEVs.

### NLRP3 inflammasome activation in response to GEVs infection in murine peritoneal macrophages

To examine inflammasome activation, the mRNA levels and protein expression levels of NLRP3, pro-capsae-1, pro-IL-1β in cell lysates and secretion of active caspase1 p20 and IL-1β p17 in supernatants (SUP) were detected under different treated time-ponits of 6 h, 12 h, 24 h or inoculation amount of 12.5 μg/mL, 25 μg/mL, 50 μg/mL using western blot. As shown in Fig 5, GEVs could induce obvious mRNA transcription and protein expression of NLRP3, capsae-1, and IL-1β. Caspase-1, a 45-KDa pro-enzyme in cell lysates, begins autoproteolysis to become the active form (p20 and p10) after recruiting to an activated inflammasome and then cleave pro-IL-1β (31 kDa) into active IL-1β (17 kDa). For the different inoculation time assays, the mRNA level of NLRP3 inflammasome-related genes increased comparing with the NC group, NLRP3 mRNA level significantly increased at 12 h vs 6 h (**$p < 0.01$) and stayed about

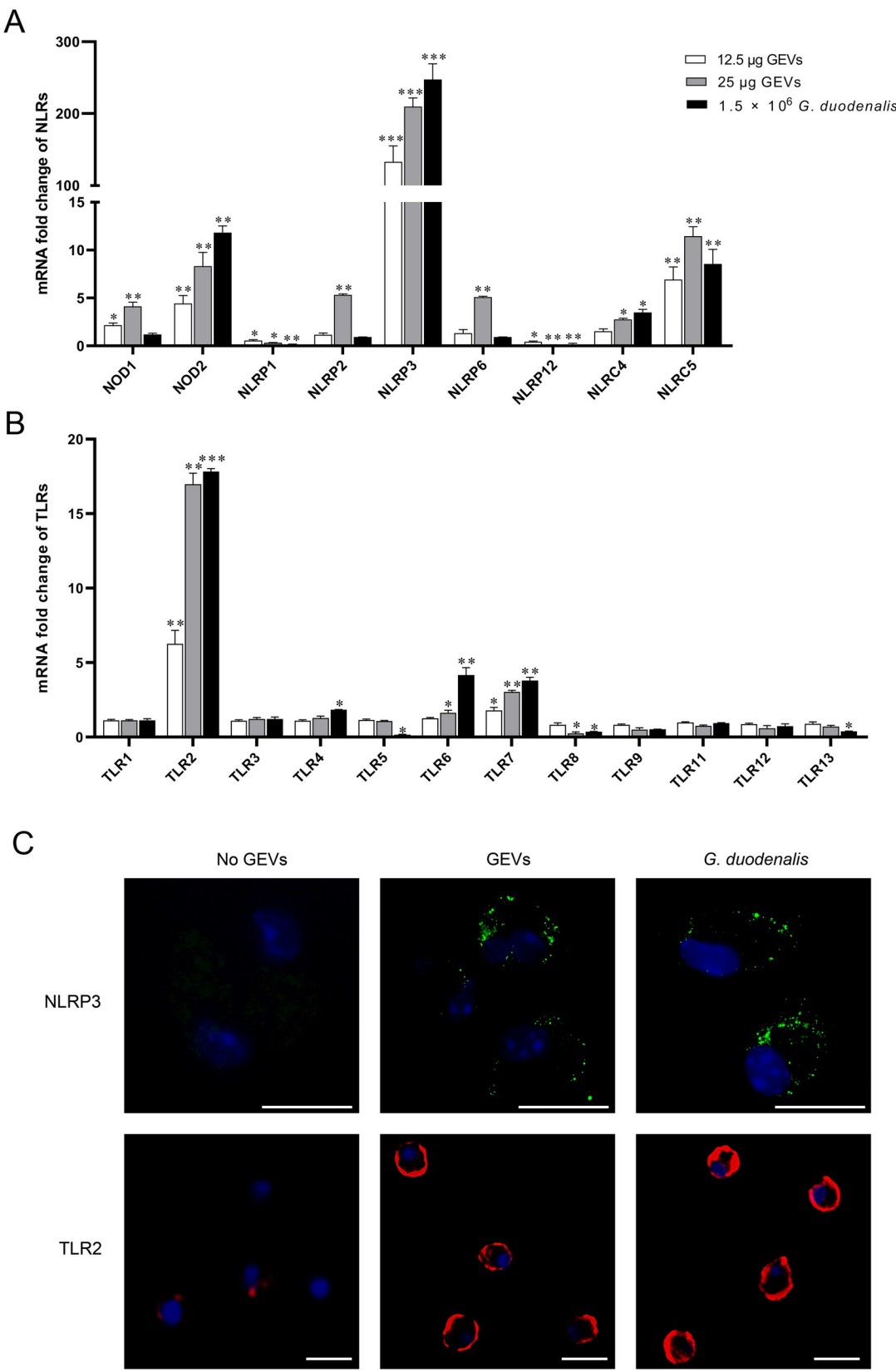

**Fig 4. Transcriptional and expression levels of pattern recognition receptors in murine peritoneal macrophages in response to GEVs infection.** (A, B) Cells coated on 12-well plate were incubated with 12.5 μg/mL or 25 μg/mL of GEVs for 12 h and the mRNA fold change of TLRs (A) and NLRs (B) in macrophages was measured using qPCR assays. Results are representative of three independent experiments with three technical replicates and data are mean±SEM. *p < 0.05, **p < 0.01 or ***p <0.001 vs. control. (C) Twenty-five microgram of GEVs were inoculated into murine peritoneal macrophages previously coated on the coverslips in the 24-well plate for 12 h and the proteins location was observed through immunofluorescence assays. *G. duodenalis* (1.5 × 10^6 parasites/mL) were used as positive control and No GEVs-treated groups were used as negative control. The green signals were FITC-labeled NLRP3 protein. The red signals were Cy3-labeled TLR2 protein. The blue signals were nuclei. Scale bars: 15 μm.

the same level at 24 h, IL-1β mRNA level shows no obvious change, and caspase1 mRNA level increased significantly along with the prolonging inoculation time ranging from 6 h to 24 h (Fig 5B). The NLRP3 inflammasome-related protein increased when comparing with NC group, in addition, the protein expression levels significantly increased at 12 h vs 6 h and then decreased at 24 h except for NLRP3, which stayed about the same level both at 6 h and 12 h (Fig 5C). For the different inoculation amount assays, the increased mRNA level of NLRP3

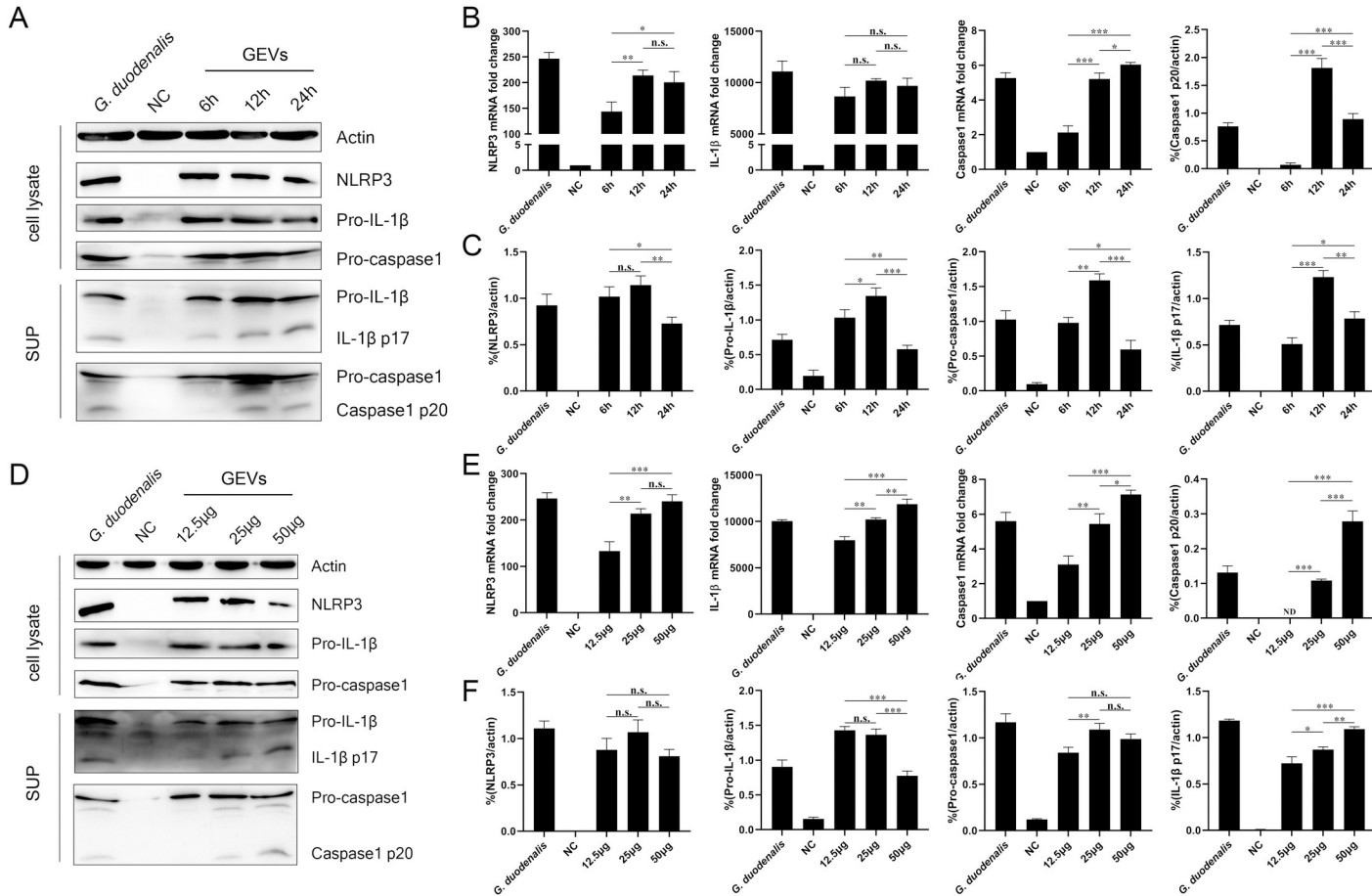

**Fig 5. Activation of NLRP3 inflammasome in murine peritoneal macrophages.** Cells were incubated with 25 μg/mL of GEVs for 6 h, 12 h and 24 h or incubated for 12 h with 12.5, 25, 50 μg/mL of GEVs. Protein expression of NLRP3 (110 KDa), Pro-IL-1β (35 KDa), Pro-caspase1(45 KDa) in the cell lysate and IL-1β p17 (17 KDa), Caspase1 p20 (20 KDa) in the SUP were detected using western blot (A and D). The mRNA fold changes of NLRP3, IL-1β, Caspase1 were detected using qPCR (B and E). The protein expression levels were measured by calculating the ratio percentage of band density in target protein and housekeeping actin. (C and F). NC represented No GEVs-treated negative control group. *G. duodenalis* in A represented positive control with 1.5 × 10^6 parasites/mL for 24 h and in D represented positive control with 1.5 × 10^6 parasites/mL for 12 h. SUP represented cell culturing supernatants. The data are expressed as the mean ± SEM from three separate experiments, *p < 0.05, **p < 0.01 or ***p <0.001 for different treated time groups or different treated amount groups.

inflammasome-related genes displayed an GEVs amount-dependent manner except for NLRP3 gene at 12.5 μg/mL and 25 μg/mL groups (Fig 5E). Moreover, there was no significant difference in the protein expression level of NLRP3 when treated with different amounts of GEVs. The pro-IL-1β protein level was about the same when treated with 12.5 or 25 μg/mL of GEVs and decreased when treated with 50 μg/mL of GEVs comparing with low amounts group (***$p < 0.001$). The pro-caspase1 protein level showed an up-regulated trend in a partially dose-dependent manner (**$p < 0.01$). The active caspase1 p20 showed a first increased and then decreased trend (**$p < 0.01$), however, the cleaved IL-1β p17 presented an increasing production (*$p < 0.05$, **$p < 0.01$ or ***$p < 0.001$) along with the increasing inoculation amount (Fig 5F). Furthermore, we also set *G. duodenalis*-treated group as control and found that *G. duodenalis* could trigger NLRP3 inflammasome activation. Taken together, these data suggest that *G. duodenalis* activates the NLRP3 inflammasome signaling pathway in murine peritoneal macrophages, probably through components in its secreted GEVs. Moreover, GEVs induce NLRP3 inflammasome activation partially depends in a time and dose manner.

## Roles of vital pattern recognition receptor TLR2 and NLRP3 in GEVs triggered inflammatory response

To further determine the relationship between activated TLR2 and NLRP3 and inflammatory response to GEVs or *G. duodenalis* infection, inhibitor assays were carried out. After pre-treatment with TLR2 inhibitor of C29, NLRP3 inhibitor of Glibenclimide (inhibiting K$^+$ efflux) or CA-074 Me (inhibiting cathepsin B) for 1 h, cells were then treated with GEVs or *G. duodenalis* and inflammatory cytokines were detected using qPCR at 12 h, ELISA at 18 h and western blot assays at 24 h. Results showed that the mRNA levels of inflammatory cytokines, including Il1β, Il6, Il10, Il17, Ifng, and Tnf were significantly down-regulated both in the GEVs-treated groups and *G. duodenalis*-treated groups after treatment with inhibitors (*$p < 0.05$, **$p < 0.01$ or ***$p < 0.001$). For Il12, Gliben or CA-074 Me treated group significantly reduced its transcription level (***$p < 0.001$), however, C-29 up-regulated its transcription (*$p < 0.05$). For Il18, Ccl20 and Cxcl2, there was no significant changes in Glibenclimide-treated groups, in contrast, the C29 or CA-074 Me groups could decrease its transcription level (*$p < 0.05$ or **$p < 0.01$) (S1 Fig). Results of ELISA assays showed that the secretion levels of IL-1β, IL-6 and TNF-α were all significantly decreased after pre-treatment with these three inhibitors (**$p < 0.01$ or ***$p < 0.001$) (Fig 6A). To further confirm the inhibition of inflammatory response, western blot assays were carried out and the protein expression levels of IL-1β p17 in the supernatants showed that CA-074 Me could completely inhibit IL-1β p17 production and C29 or Glibenclimide could partly inhibit the production of IL-1β (***$p < 0.001$) (Fig 6B). These data indicate that inhibition of activated vital pattern recognition receptor TLR2 and NLRP3 can decrease the inflammatory response in response to GEVs or *G. duodenalis* infection.

## Roles of NLRP3 inflammasome in IL-1β release and regulation of inflammatory response against GEVs infection

To determine whether NLRP3 inflammasome plays a role in GEVs-induced IL-1β release and inflammatory response, cells were pretreated with inhibitors of NLRP3 inflammasome for 1 h and then inoculated with 25 μg/mL of GEVs or $1.5 \times 10^6$ parasites/mL *G. duodenalis*. The mRNA levels of Il1β after inoculation for 12 h or IL-1β protein secretion levels after inoculation for 24 h were measured and results of qPCR and ELISA showed that NLRP3 inhibitors significantly decreased Il1β gene transcription and IL-1β protein production both in the GEVs and *G. duodenalis* groups (**$p < 0.01$ or ***$p < 0.001$) (Fig 7A and 7B). Furthermore, the

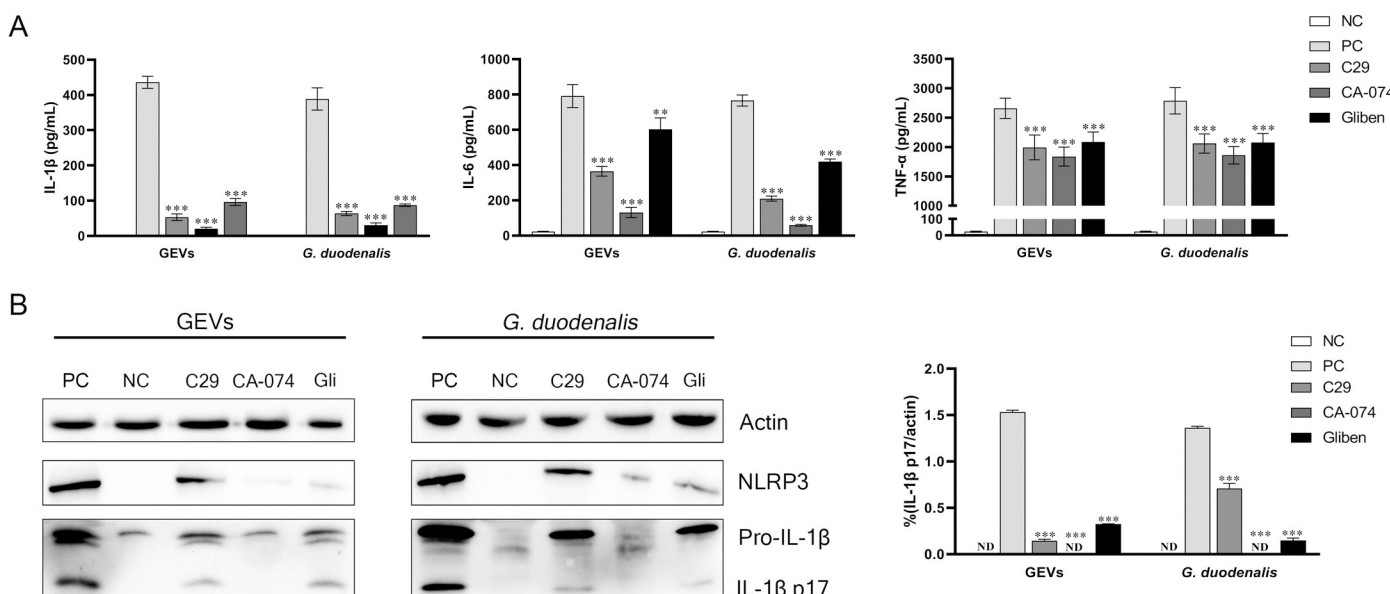

**Fig 6. Roles of TLR2 and NLRP3 receptors in GEVs triggered inflammatory response.** Murine peritoneal macrophages were pre-treated with 100 μM C29, 100 μM Glibenclimide, or 25 μM CA-074 Me for 1 h and then added 25 μg/mL GEVs or $1.5 \times 10^6$ parasites/mL *G. duodenalis* for 24 h. PBS-treated was used as negative control (NC) and GEVs or *G. duodenalis*-treated alone was used as positive control (PC). (A) Supernatants were collected and the protein expression levels of inflammatory cytokines, including IL-6, IL-1β, and TNF-α were measured using ELISA assays. (B) Protein expression level of inflammatory cytokine IL-1β in the culturing supernatants was detected using western blotting assay and densitometric percentage of IL-1β and actin in GEVs and *G. duodenalis* were calculated. Results are representative of three independent experiments with three technical replicates and data are mean±SEM. $^*p < 0.05$, $^{**}p < 0.01$ or $^{***}p < 0.001$ vs. PC of GEVs group or *G. duodenalis* treated group.

protein expression levels of IL-1β were detected using western blot and results indicated that although NLRP3 was expressed, CA-074 methyl ester or zVAD-fmk (an inhibitor of pan-caspase) comepletely inhibited the maturation of IL-1β ($^{***}p < 0.001$), in contrast, Glibenclimide or Ac-YVAD-CHO (an inhibitor of caspase-1 and -4) partially inhibited the secretion of IL-1β ($^{**}p < 0.01$ or $^*p < 0.05$) (Fig 7C and 7D). To further explore roles of activated NLRP3 inflammasome in the inflammatory response, two most obviously activated cytokines of IL-6 and TNF-α were determined via measuring the transcription levels and protein secretion levels. As shown in Fig 7E–7H, the mRNA levels of Il6 and Tnf were significantly decreased when treated with inhibitors of NLRP3 inflammasome, which displayed consistent results with that in the protein secretion levels ($^*p < 0.05$ or $^{**}p < 0.01$ or $^{***}p < 0.001$). Thus, we conclude that secretion of IL-1β in response to GEVs infection *in vitro* is mediated by NLRP3 inflammasome and NLRP3 inflammasome plays vital roles in regulating inflammatory response.

## Discussion

Diarrheal diseases occupy the majority of death and illness for children under five years old especially in developing countries. Gastrointestinal protozoan parasite of *G. duodenalis* is widely existed in the contaminated water causing waterborne diarrhea. Despite several anti-giardial drugs has been used to treat with giardiasis, however, more and more cases of treatment failure and drug resistance occurred in recent years [4,10,11]. EVs are reported to participate in the course of many diseases via delivering proteins or nucleic acids to hosts. Innate immune system is the first line against varieties of pathogens; thus, more detailed mechanisms for the roles of GEVs on the immune system are needed to be explored. The present study characterized GEVs from *G. duodenalis* and explored the effects of GEVs on the host cell innate immunity using primary mouse peritoneal macrophages and its potential mechanisms.

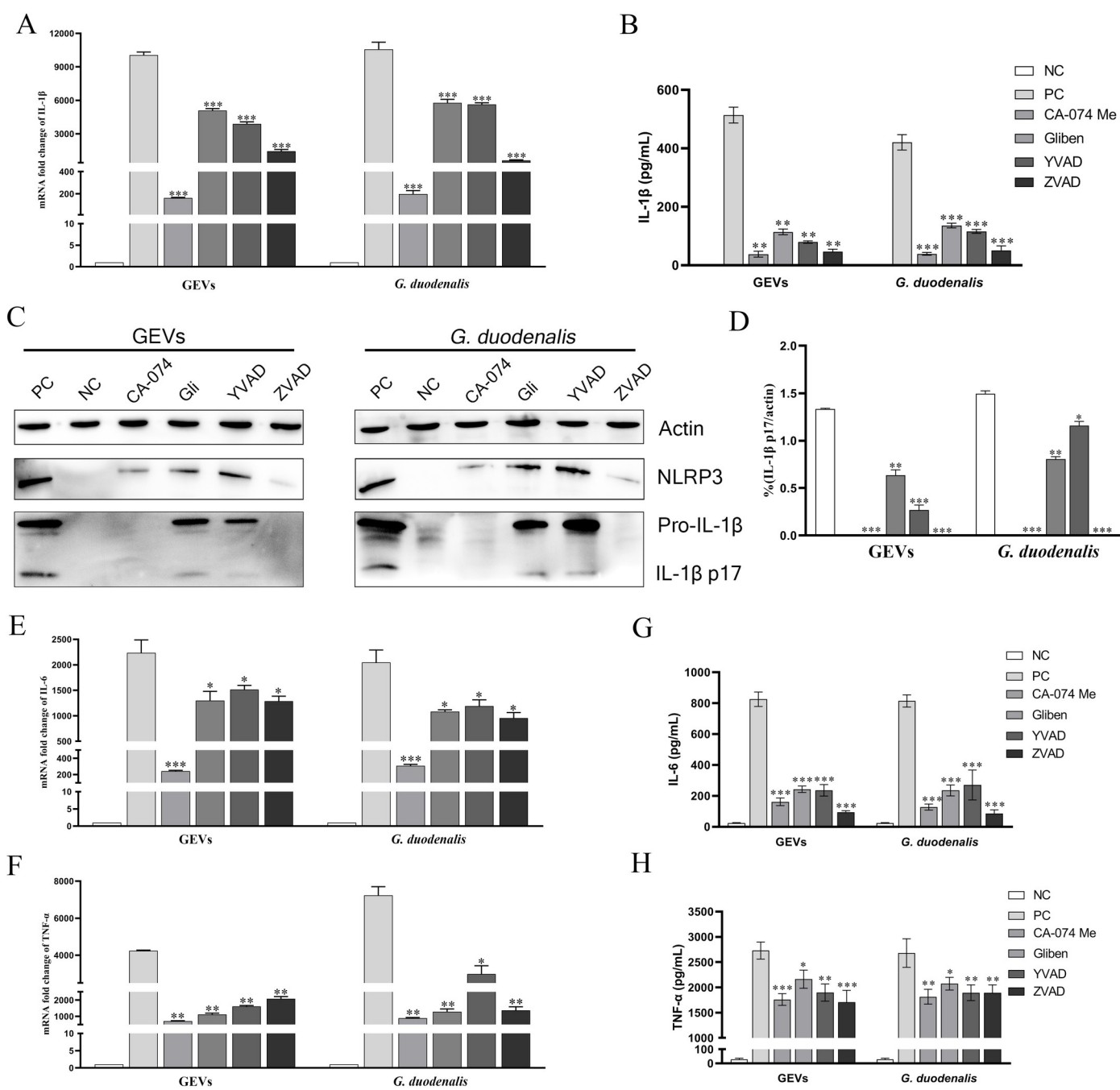

**Fig 7. Roles of NLRP3 inflammasome in IL-1β release and inflammatory response.** Murine peritoneal macrophages were pre-treated with either 50 μM Glibenclimide, 25 μM CA-074 methyl ester, 100 μM Ac-YVAD-CHO, or 10 μM zVAD-fmk for 1 h and then inoculated with 25 μg/mL GEVs or $1.5 \times 10^6$ parasites/mL *G. duodenalis* for 12 h or 24 h. The mRNA level of pro-Il1β in cells were measured (A). The protein expression levels of IL-1β p17 in the SN were detected using ELISA (B) and western blot (C). Ratio percentage of active IL-1β p17 and housekeeping actin was calculated (D). The mRNA levels of Tnf and Il6 were measured using qPCR (E-F). Secretion of TNF-α and IL-6 were measured using ELISA (G-H). The data are expressed as the mean ± SEM from three separate experiments, $^*p < 0.05$, $^{**}p < 0.01$ or $^{***}p < 0.001$ for different treated time groups or different treated amount groups.

EVs can be secreted by various cells and functioned by transmitting signals from pathogens to hosts for antigen presentation and other aspects of host defense [38]. There has been a surge of reports on EVs biological function during various disease in the past five years [39,40]. For

*G. duodenalis*, roles of EVs, including microvesicles and large EVs, were mainly focused on the attachment to intestinal epithelial cells [30,31]. Little was studied on the regulation mechanisms of GEVs induced immune response in hosts except for Evans-Osses et al. who reported that *G. duodenalis* microvesicles could increase the activation and allostimulation of human dendritic cells [31]. Considering that EVs, including microvesicles, exosomes, large and small EVs have been successfully isolated and characterized, we referred to theses EVs enrichment methods and enriched GEVs [26,31]. We show herein that GEVs were successfully prepared by the gold standard method of ultracentrifugation and presented typical round or cup-shaped structure with mean 150 nm in diameter, which was accordance with the characteristics of classical EVs [38,41]. In addtion, we compared our proteomic data with the EVs proteins in other protozoan parasites, such as *Leishmania* or *Neospora caninum*, and found that there are extensive overlaps [32,42]. Referring to these published EVs internalization assays, we evaluated the capture ability of macrophages through labeling GEVs with PKH67 dye, which could specially combine with cell membrane, and then determined the amounts of PKH67 positive cells using flow cytometry [43]. GEVs could be rapidly internalized into macrophages (19–33% PKH67$^+$ cells) within 1 h; in contrast, *G. intestinalis* microvesicles were captured by immature dendritic cells (30–40% PKH67$^+$ cells) within 30 minutes. The difference may attribute to the immune cell types, cell or GEVs amounts. Moreover, this process could be inverted by cytochalasin D, which was consistent with microvesicles in immature dendritic cells [31]. These data illustrated that GEVs could be captured by macrophages through active phagocytosis. Go annotations of GEVs showed that 95 genes, including some virulence or metabolism related factors, may involve in the immune response. Primary murine peritoneal macrophages are regarded as a typical model in immune related studies [14,19,32,44]. Moreover, macrophages have been reported to ingest *G. duodenalis* trophozoites and accumulate in the lamina propria after *G. duodenalis* invasion [45]. In this study, we isolated primary peritoneal macrophages from fluid thioglycollate medium-stimulated C57BL/6 female mice and flow cytometry showed that the purity of macrophages were 99.1%. Then, we explored whether GEVs could trigger immune response by measuring the inflammatory cytokines expression in GEVs infected macrophages. Results showed that many NF-κB-mediated cytokines or chemokines, including Il6, Tnf, and Il1β, Il18, Il12, Ifng, Ccl20 and Cxcl2, involved in the process. These data were consistent with that *G. duodenalis* trophozoites treated macrophages or *G. duodenalis* excretory-secretory products treated intestinal epithelial cells [14,36]. Meanwhile, the transcriptional level of anti-inflammatory cytokine IL-10 was also up-regulated [46,47]. This unexpected result demonstrated the unique role of GEVs in promoting sustained function of *G. duodenalis* within the host cells. Overall, *G. duodenalis* could use its secreted GEVs to manipulate immune responses similar to *Neospora caninum*, *Helicobacter pylori*, Epstein-Barr Virus, *Staphylococcus aureus* [32,48,49].

PRRs in macrophages are responsible for the recognition of PAMPs and induction of immune responses. Membrane-bound derived TLRs are widely existed in various immune cells and have been regarded as targets for therapeutic drug development in many inflammatory diseases [50]. TLRs mainly play roles in defensing against infection via expression and secretion of various pro-inflammatory cytokines induced inflammation and promoted antigen presentation or induce costimulating molecule expression and initiate specific immune response. It is reported that TLR2/p38/ERK signal pathway is essential for macrophages to induce pro-inflammatory cytokines and efficiently prime adaptive immune system against *G. duodenalis* [14]. Serradell et al. reported that *Giardia* VSP1267 could active TLR4 and TLR2 receptors after replacement of he C-terminal transmembrane region and the cytoplasmic residues with a His purification tag 6 [51]. Moreover, our proteomic data of GEVs showed that several VSPs existed in the identified proteins and KEGG analysis showed that Toll-like

receptor signaling pathway was activated. Thus, whether PAMPs of GEVs participate in TLRs-mediated pathway need to been fully elucidated. In this study, we measured twelve kinds of TLRs expression levels and explored the activated TLR's roles in regulating hosts' inflammatory responses when exposure macrophages to GEVs and *G. duodenalis*. Results indicated that many TLRs involved in this process except for TLR1, TLR3, TLR9, TLR11, and TLR12. Among them, TLR2 was the most obviously up-regulated PRR both in the GEVs and *G. duodenalis* infected macrophages. TLR2 senses the widest range of PAMPs, such as lipoproteins, lipoarabinoman-nan, glycoinositolphosp-holipids, glycolipids, porins, etc., [51] and plays important roles in reg-ulating immune responses [52,53]. Results of inhibition assays demonstrated that cytokines of Il1β, Il6, Il10, Il17, Ifng, Tnf, Il18, Ccl20 and Cxcl2 were significantly down-regulated not only in the *G. duodenalis*-inoculated but also in the GEVs-inoculated macrophages after pretreat-ment with specific TLR2 inhibitor of C29 [54]. The mRNA level of Il12 gene was up-regulated in the C29-pretreatment groups. To further verify this result, we measured the IL-12 p40 secre-tion level using ELISA and found that the average level of IL-12 after pretreatment with C29 was 24.085 pg/mL, which was higher than that in the control group (8.425 pg/mL). Similar to our study, Obendorf et al. showed that cytokines of IL-12, IL-23, and IL-10 were enhanced in *G. duodenalis* inoculated human dendritic cells in the presence of TLR2 ligands [55]. Li et al. found that cytokines of TNF-α, IL-6, and IL-12 p40 was increased in *G. duodenalis* infected TLR2 deficiency or TLR2 antibody neutralization murine macrophages, which were adverse with our results [14]. This might probably due to the different choices of inoculation ways or infection dose. Overall, we conclude that TLR2 can be distinctly activated when exposure mac-rophages to GEVs and *G. duodenalis* and some PAMPs that can trigger TLR2 are present in the GEVs. In addition, both GEVs and *G. duodenalis* can modulate varieties of cytokines expression in macrophages with a manner dependent on TLR2.

NLRs are mainly expressed in immune cells and shares the function of NF-κB activation, such as NOD1 and NOD2, or secretion of pro-inflammatory cytokines IL-1β and IL-18, such as NLRP3 [56]. NLRP3 has been deeply studied since it could recognize various bacteria, viruses, fungi, parasites and nigericin and uric acid crystals, etc. and involves in many intracellular path-ogens induced disease, such as *Staphylococcus aureus* bacteremia [17,57,58]. However, limited research focus on the immune mechanisms mediated by the intracellular NLRs in extracellular pathogen of *G. duodenalis* except for a newly research, which reports that *G. duodenalis* can attenuate giardiasis *in vivo* via NLRP3 [22]. Moreover, the exact PAMPs that deliver biological information into host cells still need to be determined. In the present study, we detected varie-ties of NLRs expression levels when exposure macrophages to GEVs and *G. duodenalis* and found that NOD2, NLRP3, NLRC4 and NLRC5 were obviously up-regulated. Among them, NLRP3 was the most obviously up-regulated receptor. Furthermore, many cytokines were inhibited when NLRP3 was blocked with either Glibenclimide or CA-074 methyl ester [59,60]. Changes of NLRP3 and IL-1β levels led us to explore the process of IL-1β secretion and NLRP3 inflammasome activation. We inoculated GEVs into macrophages for different time and with different dose and then detected the expression levels of NLRP3, pro-IL-1β, pro-caspase-1, active IL-1β p17, and active caspase-1 p20. *G. duodenalis*-treated group was used as positive control and no treatment group was used as negative control. Results indicated that NLRP3 inflammasome was activated and IL-1β production was in dose-dependent and time dependent manners, which was accordance with *Naegleria fowleri* or *N. caninum in vitro* and *G. duodenalis in vivo* [22,34,59]. To further determine the formation ways of GEVs-triggerred NLRP3 inflam-masome and its role in IL-1β release, Glibenclimide (an inhibitor of NLRP3 inflammasome via blocking K⁺ efflux), CA-074 methyl ester (an inhibitor of NLRP3 inflammasome via blocking cathepsin B), Ac-YVAD-CHO (an inhibitor of NLRP3 inflammasome via blocking caspase-1 and -4) or zVAD-fmk (an inhibitor of pan-caspase) were pretreated with macrophages and the

expression levels of IL-1β was detected [19,59,61]. Results showed that little IL-1β secretion was detected when treated with CA-074 methyl ester or zVAD-fmk. In contrast, the IL-1β secretion was partly inhibited in the presence of Glibenclimide or Ac-YVAD-CHO. These data illustrated that NLRP3 inflammasome activation was mainly via cathepsin B way and the formation of NLRP3 inflammasome was not only dependent on the canonical pathway but also dependent on non- canonical signaling pathway [62]. Furthermore, we found that block NLRP3 inflammasome activation down-regulated inflammatory cytokines levels, which was coinstent with that in Bisphenol S-induced inflammation in mrine RAW264.7 cells via NLRP3 inflammasome [63]. These data uncovered an intriguing mechanism that GEVs secreted by *G. duodenalis* triggered the intracellular NLRP3 inflammasome activation and IL-1β secretion, which played a crucial role in regulating host's inflammatory response.

In conclusion, we illustrate that GEVs secreted by *G. duodenalis* are actively involved in the inflammatory response in primary murine peritoneal macrophages by activation TLR2 and NLRP3 inflammasome signaling pathways. Moreover, GEVs induced NLRP3 inflammasome in macrophages mediate caspase-1 processing and IL-1β secretion, which consequently causes inflammation (Fig 8). These findings will be important to elucidate the immunotoxicity of

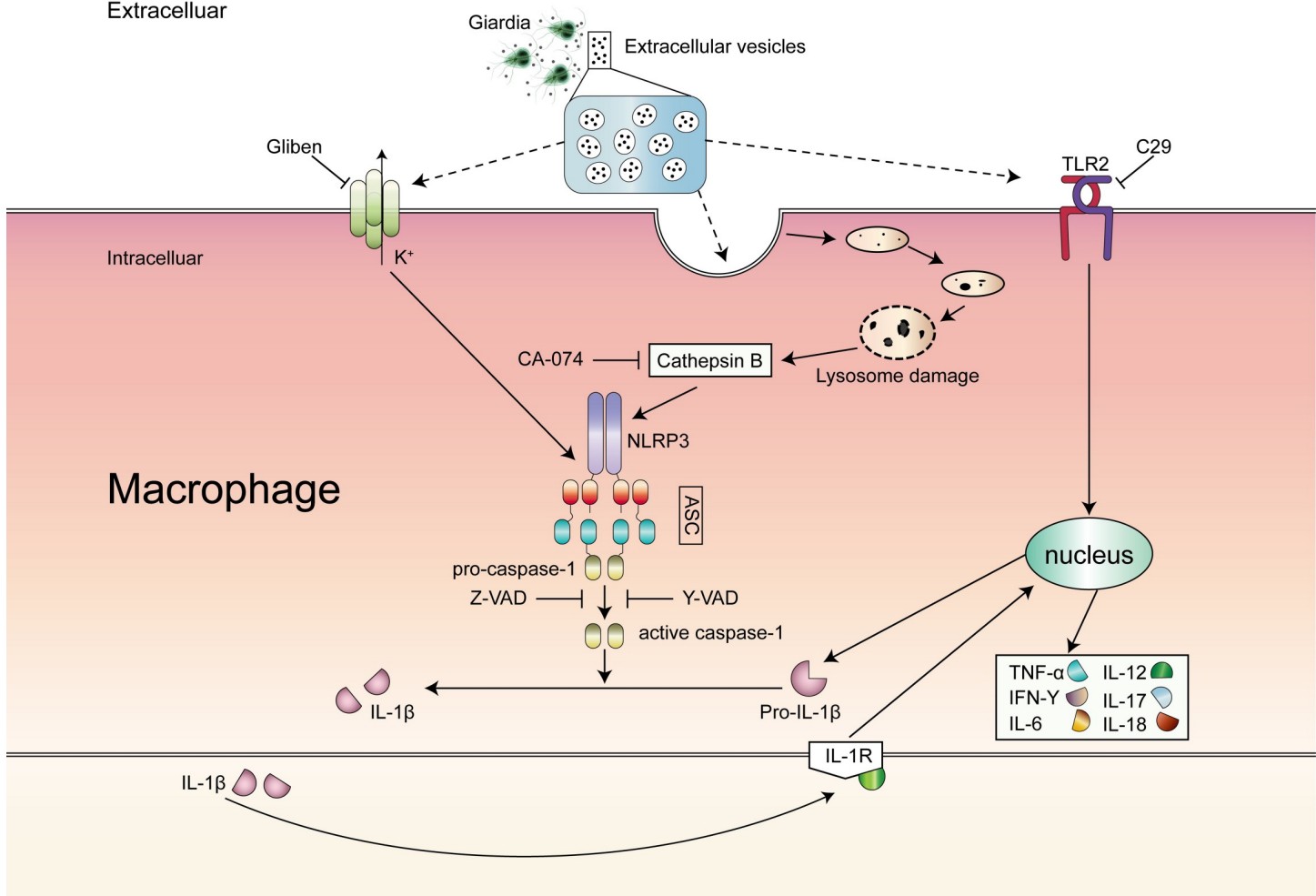

**Fig 8. Schematic diagram of mechanisms underlying GEVs-induced inflammatory response.** GEVs were secreted by *G. duodenalis* and entered into host cells triggering inflammatory response by recognizing extracellular TLR2 and intracellular NLRP3. NLRP3 inflammasome was activated, mediated IL-1β release and controlled cytokines secretion. This process could be completely reversed by CA-074 methyl ester/zVAD-fmk or partial reversed by C29/Glibenclimide/Ac-YVAD-CHO.

GEVs *in vivo* in future studies and lay a foundation for looking for new targets against giardiasis.

## Supporting information

**S1 Fig. Roles of TLR2 and NLRP3 receptors in GEVs triggered cytokines transcriptional levels.** Murine peritoneal macrophages were pre-treated with 100 μM C29, 100 μM Glibenclimide, or 25 μM CA-074 Me for 1 h and then added 25 μg/mL GEVs or $1.5 \times 10^6$ parasites/mL *G. duodenalis* for 12 h. PBS-treated was used as negative control. Cells were collected and the mRNA expression levels of cytokines were measured using qPCR assays.
(TIF)

**S1 Table. All proteins identified in the GEVs secreted by *G. intestinalis*.**
(XLSX)

**S2 Table. Go annotations of GEVs.**
(XLS)

**S3 Table. KEGG analysis of GEVs.**
(XLS)

## Author Contributions

**Conceptualization:** Panpan Zhao, Lili Cao, Xiaocen Wang, Xichen Zhang, Pengtao Gong.

**Data curation:** Panpan Zhao, Jingquan Dong.

**Formal analysis:** Panpan Zhao, Lili Cao, Xiaocen Wang, Jingquan Dong, Xin Li, Xichen Zhang, Pengtao Gong.

**Funding acquisition:** Pengtao Gong.

**Investigation:** Panpan Zhao, Lili Cao.

**Methodology:** Panpan Zhao, Lili Cao, Xiaocen Wang, Jingquan Dong, Nan Zhang, Xin Li, Pengtao Gong.

**Project administration:** Xichen Zhang, Pengtao Gong.

**Resources:** Xin Li, Jianhua Li, Xichen Zhang, Pengtao Gong.

**Software:** Panpan Zhao.

**Supervision:** Nan Zhang, Jianhua Li.

**Validation:** Jingquan Dong, Nan Zhang, Jianhua Li, Pengtao Gong.

**Visualization:** Nan Zhang, Xin Li, Jianhua Li, Xichen Zhang.

**Writing – original draft:** Panpan Zhao.

**Writing – review & editing:** Pengtao Gong.

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
