## [Decision Letter · Decision Letter 0]

28 Dec 2020

Dear Dr. Gong,

Thank you very much for submitting your manuscript "Extracellular vesicles secreted by Giardia duodenalis regulate host cell innate immunity via TLR2 and NLRP3 inflammasome signaling pathways" for consideration at PLOS Neglected Tropical Diseases. As with all papers reviewed by the journal, your manuscript was reviewed by members of the editorial board and by several independent reviewers. In light of the reviews (below this email), we would like to invite the resubmission of a significantly-revised version that takes into account the reviewers' comments. 

The message of the manuscript is timely and relevant, but there are major concerns from the 3 reviewers regarding the format/writing, appropriate citations and additional specific questions raised by the reviewers that needs to be addressed.

We cannot make any decision about publication until we have seen the revised manuscript and your response to the reviewers' comments. Your revised manuscript is also likely to be sent to reviewers for further evaluation.

Sincerely,

Dario S. Zamboni, Ph.D.

Associate Editor

Steven Singer

Deputy Editor

The message of the manuscript is timely and relevant, but there are major concerns from the 3 reviewers regarding the format/writing, appropriate citations and additional specific questions raised by the reviewers that needs to be addressed.

Reviewer's Responses to Questions

**Key Review Criteria Required for Acceptance?**

**Methods**

-Are the objectives of the study clearly articulated with a clear testable hypothesis stated?

-Is the study design appropriate to address the stated objectives?

-Is the population clearly described and appropriate for the hypothesis being tested?

-Is the sample size sufficient to ensure adequate power to address the hypothesis being tested?

-Were correct statistical analysis used to support conclusions?

-Are there concerns about ethical or regulatory requirements being met?

Reviewer #1: my concern for authors not to be guided or cite important literature in the area. Experiments with macrophages and internalization must be improverd. Authors do not cite and do not know essential works in the area

Reviewer #2: Line 330: LAL assay measures LPS which is not present in Giardia EVs. Also, the relevance of LAL values should be mentioned in the discussion not in the results section. LAL is not the right method here as it measures LPS in gram negative bacteria. 

Lines 331: 25 ug/mL Did the authors do a BCA to quantify protein content? How did the authors isolate 25 ug/mL? Is it a physiological concentration? A range of concentrations should be used. 

Line 336: Did the authors use a positive control? How about Giardia trophozoites?

In the methods section: The proteomic content of EVs was not analyzed. The authors should further characterize EVs protein content. (See Thery et al. article in JEV for guidelines about EVs research). 

Figure 4, 5, 6, 7, 8, 9: Do the data illustrate “fold change” or “ratios”? No SEM is indicated in non treated groups. How did the authors do statistical analysis with Ratios? 

Lines 31-32. Please specify the target selectivity and effects of the inhibitors.

Line 38: anti- giardiasis should be reworded as anti-Giardia

Line 50-52. Please rephrase these lines. This is not the first study to demonstrate the internalization of Giardia EVs. Refer to Evans Osses et al. 2017. 

Line 86-88: Please rephrase. There is canonical and non-canonical NLRP3 activation.

Line 90. A reference is needed for the statement regarding the first signal in the stimulation of NLRP3. 

Lines 100-102: Please rephrase.

Line 103. Please remove “of” before the names of the parasites. Same for Line 105.

Line 109-113: Please rephrase. Also introduce the term GEVs before abbreviating it. 

Line 111: “Invested” should be replaced with “investigated”. 

Line 127 : How is an anaerobic environment created? Also, the term bottles is confusing. Were tubes used? Please note that Giardia is microaerophilic. 

Line 132: The protocol cited for the isolation of EVs does not refer to Giardia EVs isolation. 

Line 134: Add a reference of the product. (Exosome depleted FBS).

Line 150: The term tachyzoites (Toxoplasma) is used. Instead, it should be trophozoites.

Line 157: Can you please describe the NTA protocol in detail like what camera settings, threshold etc. were used.

Line 165-169: The protocol for labelling EVs using PKH-67 is based on Evans-Osses et al. and it should be cited. 

Line 188: Please indicate the time point for stimulation of macrophages. 

Line 193: Please describe the protocol for macrophages/Giardia Transwell. Which media was used? Note that Giardia can stress the cells. 

Line 217 (Table 1) Please follow the nomenclature for murine genes (ex: Il10 instead of IL10).

Lines 234: Please change the title. (Inflammasome molecules should be removed.)

Lines 276-279 (Statistical analysis section) : How was the normality of the data assessed? What kind of post-tests have been used? No T-test is used? Was the data parametric or non-parametric. 

Line 282: Please rephrase the title of the section. This has been shown and characterized in previous studies. 

Line 289: Can you indicate the diameter (mean) on Figure 1D. 

Line 290: Figure 1C should be removed. NTA images are not relevant.

Line 390: Is Glibenclimide NLRP3-selective?

Line 391: CA-074-Me is not a direct inhibitor of NLRP3.

Line 428: Why did the authors not use different concentration of EVs before? i.e. in figures 4,5 and 6.

Line 431: The word “platform” does not fit here. 

General comment: The authors use ratio instead of doing to fold change and compare it to control. Also, no SD/SEM bars on the control Please edit. 

Line 477: zVAD-fmk: Please introduce this term in this section. It inhibits caspase at the first place. 

Lines 484-486: Increase of IL-1 beta is not observed in vivo in mice infected with G. duodenalis. 

Lines 500-504: Move to introduction. 

Lines 508-510: Rephrase. 

Lines 525-526: The cytokines increased are NF-kB cytokines. 

Lines 527: We can’t really conclude about Th1/Th2 balance as the cytokines are observed in macrophages. Have the authors used NF-kB inhibitors?

Line 521: What is the role of IL-10? Pro-resolving

Lines 528-533: How about the apoptosis and necropsy assays? 

Line 550: “Immune inflammatory response” could be reworded. 

Lines 553-561: Please adjust the wording and rephrase. 

Line 588: Authors also need to cite Manko et al. 2020 (International journal of parasitology)

Lines 590. Glyburide NLRP3 inhibitor is commonly used to inhibit NLRP3 inflammasome activation. Why did Glyburide not used in this study?

Line 603 “environmental hazardous” should be reworded.

Line 605-607: Activation of NLRP3 was first demonstrated by Manko et al IJP 2020. The novel findings is the role of EVs.

Reviewer #3: Line 127. Is it bottles or tubes? Was it anaerobic or microaerophilic?

Line 133. What volumes/number of cells were used in the preparations of EVs?

Line 145. Giardia has no tachyzoite stage, it is trophozoites! This is elementary knowledge and the same mistake is repeated through-out the paper. 

Line 160. How pure is this cell population? Are there other immune cells like mast cells, DCs, neutrophils?

**Results**

-Does the analysis presented match the analysis plan?

-Are the results clearly and completely presented?

-Are the figures (Tables, Images) of sufficient quality for clarity?

Reviewer #1: The Ev characterizathion and internalization into macrophages are not clear.

Reviewer #2: Line 287: Figure 1A and Figure 1b : EVs can not be clearly identified in the images. New images are required to confirm 1) EVs 2) Lipid Bilayer. 

Lines 291 and 292: SEM images indicate that the trophozoites are not healthy or even viable, or may have been damaged in the SEM preparation process. New SEM images are required. Overall, SEM images don’t show evidence of EVs and TEM images might be showing debris.

Line 318: Do GEVs induce apoptosis or necropsy? Apoptosis could explain cytokine status, Macrophage engulfment of particles is common.

Lines 296: General Comment: Figure legends should be at the end of the manuscripts (please amend in all the manuscript).

Line 334: Follow the nomenclature of the genes. 

Lines 340-345 Please rephrase. 

Line 344- “Obviously” should not be used. 

Line 361- Have the authors tried different concentrations? 

Line 363: Giardia might be causing apoptosis and necrosis of macrophages. 

Line 364: Change wording. (“Extremely obvious increased” does not make sense)

Line 366: Is TLR4 activated by Giardia?

General comment = Please do not indicate mean and SD on the histograms. P vale could be added in the figure caption instead of the figure. 

Line 403-404: Please adjust the wording.

Line 405. The term “secretion” is not correct here. Also, there is no inhibition of cleavage of pro-IL-1 beta (Blockage of NLRP3� cleavage of pro IL-1 beta to IL-1b). 

Line 421: What is the positive control? G. duodenalis? Please specify. 

Line 421: Figure 6 should be split into 2 figures as the message is confusing.

Line 420: caspase p20 should be caspase 1?

Figure 7 could be presented before figure 6. Figure 7 displays NLRP3 inflammasome activation and figure 6 uses inhibitors (mechanism)

Lines 445-446: Please rephrase. 

Line 782 (Figure 6 panel F). What is NC? Why NC is in GEV and G. duodenalis experiments? What is C? Positive control? 

Lune 784 (Figure 7) Panel A: Please specify that 6-, 12- and 24-hours treatments are for EVs. Also include size. 

Panel C&F : Specify band density in the legend. Panel D: Specify that 12.5, 25 and 50ug are concentrations of EVs for clarity. 

Also, please use cell lysate instead of LYS and SUP instead of SN. 

Line 788 (Figure 9). What is the line (the one before the conversion of IL-1beta to IL-1R) representing?

Reviewer #3: Line 108. There are several publications dealing with Giardia EVs and also innate immune responses that are missing in the paper. They should be added both to the Introduction, Results and Discussion. See these papers: 

Microvesicles released from Giardia intestinalis disturb host-pathogen response in vitro.

Evans-Osses I, Mojoli A, Monguió-Tortajada M, Marcilla A, Aran V, Amorim M, Inal J, Borràs FE, Ramirez MI. Eur J Cell Biol. 2017 Mar;96(2):131-142. doi: 10.1016/j.ejcb.2017.01.005. Epub 2017 Jan 22. PMID: 28236495

Characterization of the Giardia intestinalis secretome during interaction with human intestinal epithelial cells: The impact on host cells.

Ma'ayeh SY, Liu J, Peirasmaki D, Hörnaeus K, Bergström Lind S, Grabherr M, Bergquist J, Svärd SG. PLoS Negl Trop Dis. 2017 Dec 11;11(12):e0006120. doi: 10.1371/journal.pntd.0006120. eCollection 2017 Dec.

Peptidylarginine Deiminase Inhibition Abolishes the Production of Large Extracellular Vesicles From Giardia intestinalis, Affecting Host-Pathogen Interactions by Hindering Adhesion to Host Cells.

Gavinho B, Sabatke B, Feijoli V, Rossi IV, da Silva JM, Evans-Osses I, Palmisano G, Lange S, Ramirez MI.Front Cell Infect Microbiol. 2020 Sep 23;10:417. doi: 10.3389/fcimb.2020.00417. eCollection 2020.PMID: 33072615

Exosome Biogenesis in the Protozoa Parasite Giardia lamblia: A Model of Reduced Interorganellar Crosstalk.

Moyano S, Musso J, Feliziani C, Zamponi N, Frontera LS, Ropolo AS, Lanfredi-Rangel A, Lalle M, Touz M.Cells. 2019 Dec 9;8(12):1600. doi: 10.3390/cells8121600

10. Line 286. The quality is poor in the EMs, compare eg with the images from the references above. I do not see a double membrane in the pictures provided and there are particles of different sizes.

11. Line 290. What is shown in the C panel?

12. Line 291. This is not convincing, one particle like structure from one cell, how do the authors know that the SEM is picking up EVs in this picture?

13. Line 294. Several papers have shown this already.

14. Line 309. I would be good to see just labelled EVs in super resolution.

15. Line 316. Can co-staining be done to see in what compartment the signal from PKH67 is seen?

16. Line 319. How many macrophages were given 25ug EVs?

17. Line 326. Also secreted proteins (including EVs) induce chemokine expression in intestinal epithelial cells, see ref by Maayeh et al., 2017.

18. Line 330. Is it certain that it cannot trigger any immune response?

19. Line 333. Why were not CCL20, CXCL 1-3 and CCL2 tested since they are induced in intestinal epithelial cells according to Ref 32? Addition of this would strengthen the paper and connect it to earlier data.

20. Line 334. Delete an extremely 

21. Line 364. Delete all extremely obvious increased. Here it would be nice to test fixed trophozoites to see if they can induce a response.

22. Line 374 and 377. Up-regulated does not mean that the receptors are activated.

23. Line 388. It would be nice to see localization of TLR2 and NLRP3 in relationship to the ingested GEVs.

**Conclusions**

-Are the conclusions supported by the data presented?

-Are the limitations of analysis clearly described?

-Do the authors discuss how these data can be helpful to advance our understanding of the topic under study?

-Is public health relevance addressed?

Reviewer #1: The experiments must be done with kinetics and dose dependence assays

Reviewer #2: In the introduction section, the authors do not cite or discuss work done by other groups in the field of Giardia EVs. It would have been nice if they would have mentioned what other groups have shown regarding Giardia EVs and their involvement in various aspects of host parasite interactions including their effects on immune cells. For example: Evans-Osses et al. 2017 have shown that EVs can modulate dendritic cells. 

Line 18 (Abstract): Giardia duodenalis is written in brackets as (G. duodenalis) which is incorrect. 

Line 39: Is the author summary required for Plos Neglected tropical diseases. If not, this section can be moved to the introduction.

Line 55. Remove G. duodenalis in the brackets and instead of saying “also known”, please say “syn. G. lamblia, G. intestinalis”. Also, please note that only Assemblages A and B are zoonotic. 

Line 60. Public health relevance: Giardia is not the main cause of Traveller’s diarrhea. Think of the many other common enteropathogens, such as EPEC.

Line 61- References 6 and 7 are not the right references. 

Line 66- A reference is needed for the statement that increasing resistance to drugs is becoming common.

Overall, the current article lacks some information which would have made it more compelling, and the conclusions better supported (eg. better imaging, proteomics, use of other inhibitors). The article does not include recent findings directly relevant to this study.

Methods and results descriptions must be improved, as above.

Reviewer #3: It is very important to use the correct references in order to make good conclusions.

Line 517. Reference earlier EV work in the discussion.

Line 529. Strange sentence.

Line 558. Delete sentence, does not make sense.

Line 564. The results are very different from the two studies. Why is this? Are there differences in parasite and mouse strains, GEVs vs trophozoites or what?

Line 568. See paper by Serradell showing that VSPs induce TLR2 and VSPs have been seen in EVs during proteomics (see earlier papers above). 

Efficient oral vaccination by bioengineering virus-like particles with protozoan surface proteins.

Serradell MC, Rupil LL, Martino RA, Prucca CG, Carranza PG, Saura A, Fernández EA, Gargantini PR, Tenaglia AH, Petiti JP, Tonelli RR, Reinoso-Vizcaino N, Echenique J, Berod L, Piaggio E, Bellier B, Sparwasser T, Klatzmann D, Luján HD. Nat Commun. 2019 Jan 21;10(1):361. doi: 10.1038/s41467-018-08265-9

**Editorial and Data Presentation Modifications?**

Reviewer #1: (No Response)

Reviewer #2: I read with interest the research article by Zhao and colleagues dealing with the effects of Giardia EVs. The article is timely and investigates an important area of research in the field. The authors studied whether and how Giardia EVs may modulate the immune system. The investigators show that Giardia EVs (GEVs) may induce an immune/inflammatory response by increasing the production of cytokines. The effect was associated with the activation of TLR2 and NLRP3 inflammasome signaling pathways. The effects were diminished when macrophages were pre-treated with activation of TLR2 CA-074 methyl ester or zVAD-fmk. Overall, the current article lacks some information which would have made it more compelling.

Reviewer #3: 1. The English has to be improved through-out the paper, there quality is relatively low now, starting already in the abstract. 

2. Line 22. Explain EV and introduce GEV here.

3. Line. Remove remarkably. 

4. Line 34. “Provide new targets against giardiasis” is not necessary.

**Summary and General Comments**

Reviewer #1: the authors do not cite and do not know essential works in the area of Giardia and EVS. They do not consider manuscripts in giardiasis immunology. The authors must define experiments with two doses to know about dose dependence effect.

Reviewer #2: as indicated above.

Reviewer #3: Line 286. The quality is poor in the EMs, compare eg with the images from the references above. I do not see a double membrane in the pictures provided and there are particles of different sizes. Show enter examples.

Line 291. This is not convincing, one particle like structure from one cell, how do the authors know that the SEM is picking up EVs in this picture? Show more examples.

Line 309. I would be good to see just labelled EVs in super resolution microscopy.

Line 316. Can co-staining be done to see in what compartment the signal from PKH67 is seen?

Line 333. Why were not CCL20, CXCL 1-3 and CCL2 tested since they are induced in intestinal epithelial cells according to Ref 32? Addition of this would strengthen the paper and connect it to earlier data.

Line 388. It would be nice to see localization of TLR2 and NLRP3 in relationship to the ingested GEVs.

PLOS authors have the option to publish the peer review history of their article (what does this mean?). If published, this will include your full peer review and any attached files.

Reviewer #1: No

Reviewer #2: No

Reviewer #3: No
---

## [Decision Letter · Decision Letter 1]

10 Mar 2021

Dear Dr. Gong,

We are pleased to inform you that your manuscript 'Extracellular vesicles secreted by Giardia duodenalis regulate host cell innate immunity via TLR2 and NLRP3 inflammasome signaling pathways' has been provisionally accepted for publication in PLOS Neglected Tropical Diseases.

Best regards,

Dario S. Zamboni, Ph.D.

Associate Editor

Steven Singer

Deputy Editor

The manuscript has been significantly improved, as indicated in the reviewers' comments. Please address (and adjust accordingly) the reviewer #1 comment on the NTA experiment.

Reviewer's Responses to Questions

**Key Review Criteria Required for Acceptance?**

**Methods**

-Are the objectives of the study clearly articulated with a clear testable hypothesis stated?

-Is the study design appropriate to address the stated objectives?

-Is the population clearly described and appropriate for the hypothesis being tested?

-Is the sample size sufficient to ensure adequate power to address the hypothesis being tested?

-Were correct statistical analysis used to support conclusions?

-Are there concerns about ethical or regulatory requirements being met?

Reviewer #1: (No Response)

Reviewer #2: All ok, as revised.

Reviewer #3: Required changes have been performed.

**Results**

-Does the analysis presented match the analysis plan?

-Are the results clearly and completely presented?

-Are the figures (Tables, Images) of sufficient quality for clarity?

Reviewer #1: (No Response)

Reviewer #2: all ok as revised.

Reviewer #3: Required modifications have been made.

**Conclusions**

-Are the conclusions supported by the data presented?

-Are the limitations of analysis clearly described?

-Do the authors discuss how these data can be helpful to advance our understanding of the topic under study?

-Is public health relevance addressed?

Reviewer #1: (No Response)

Reviewer #2: all ok as revised.

Reviewer #3: (No Response)

**Editorial and Data Presentation Modifications?**

Reviewer #1: (No Response)

Reviewer #2: all ok as revised.

Reviewer #3: (No Response)

**Summary and General Comments**

Reviewer #1: The authors have made a great effort to incorporate important literature into the area and completely forgotten in the first version. Experimentally in several experiments they performed dose dependence to demonstrate effects of GEVs in interacting with cells and validate the results. Similarly it is recommended to make an editing of the text mainly in the quality of the figures. Many of the experiments could have used unrelated EVs and not just the absence of vesicles. I am not yet convinced by the result of NTA because the authors seem to have read for a minute and usually 3 one-minute readings are made. This demostrate the homogeneity of the vesicle peack shown by the authors. Due to the large number of vesicles used, the authors should compare their production in relationship to an equivalence of production for each parasite. How much the number of vesicles represents.

Reviewer #2: Authors have addressed ALL reviewers' comments (general comments, minor edits) and have performed new experiments (or included new supplementary materials); among those proteomics data, different concentrations of EVs (i.e. 12.5 ug in addition to the existing 25 ug). They have removed the LAL assay as we had suggested and removed the SEM images which is a good thing. They have updated the TEM images (we can clearly identify the membrane bilayer now) and NTA figures. They have changed figure order and improved the methods as some protocols were confusing. An impressive revision.

Reviewer #3: The paper has been modified according to the comments of the reviewers and it can now be accepted.

PLOS authors have the option to publish the peer review history of their article (what does this mean?). If published, this will include your full peer review and any attached files.

Reviewer #1: No

Reviewer #2: **Yes: **Andre G. Buret, Professor, University of Calgary (Canada)

Reviewer #3: No

---

## [Editor Report · Acceptance letter]

29 Mar 2021

Dear Dr. Gong,

We are delighted to inform you that your manuscript, "Extracellular vesicles secreted by Giardia duodenalis regulate host cell innate immunity via TLR2 and NLRP3 inflammasome signaling pathways," has been formally accepted for publication in PLOS Neglected Tropical Diseases.

Best regards,

Shaden Kamhawi

co-Editor-in-Chief

Paul Brindley

co-Editor-in-Chief
